# Kin discrimination in social yeast is mediated by cell surface receptors of the Flo11 adhesin family

**Stefan Brückner[1†], Rajib Schubert[2†‡], Timo Kraushaar[3], Raimo Hartmann[4], Daniel Hoffmann[1], Eric Jelli[4], Knut Drescher[4], Daniel J Müller[2]\*, Lars Oliver Essen[3,5]\*, Hans-Ulrich Mösch[1,5]\***

[1]Department of Genetics, Philipps-Universität Marburg, Marburg, Germany; [2]Department of Biosystems Science and Engineering, ETH Zürich, Basel, Switzerland; [3]Department of Biochemistry, Philipps-Universität Marburg, Marburg, Germany; [4]Max Planck Institute for Terrestrial Microbiology, Marburg, Germany; [5]LOEWE Center for Synthetic Microbiology, Philipps-Universität Marburg, Marburg, Germany

**\*For correspondence:**
daniel.mueller@bsse.ethz.ch (DJMü);
essenl@staff.uni-marburg.de (LOE);
moesch@staff.uni-marburg.de (H-UMö)

[†]These authors contributed equally to this work

**Present address:** [‡]Roche Sequencing Solutions, Santa Clara, California, United States

**Competing interests:** The authors declare that no competing interests exist.

**Abstract** Microorganisms have evolved specific cell surface molecules that enable discrimination between cells from the same and from a different kind. Here, we investigate the role of Flo11-type cell surface adhesins from social yeasts in kin discrimination. We measure the adhesion forces mediated by Flo11A-type domains using single-cell force spectroscopy, quantify Flo11A-based cell aggregation in populations and determine the Flo11A-dependent segregation of competing yeast strains in biofilms. We find that Flo11A domains from diverse yeast species confer remarkably strong adhesion forces by establishing homotypic interactions between single cells, leading to efficient cell aggregation and biofilm formation in homogenous populations. Heterotypic interactions between Flo11A domains from different yeast species or *Saccharomyces cerevisiae* strains confer weak adhesive forces and lead to efficient strain segregation in heterogenous populations, indicating that in social yeasts Flo11A-mediated cell adhesion is a major mechanism for kin discrimination at species and sub-species levels. These findings, together with our structure and mutation analysis of selected Flo11A domains, provide a rationale of how cell surface receptors have evolved in microorganisms to mediate kin discrimination.

## Introduction

For organisms living in communities, the ability to discriminate between kin and non-kin is fundamental for enabling close relatives to form groups with beneficial social interactions, and for preventing invasion of these groups by social cheaters (*Bonner, 2004*; *Michod, 2007*; *Strassmann and Queller, 2011*; *Wall, 2016*). Genetic systems for kin discrimination have evolved in vertebrates (*Jiang and Chess, 2009*; *Beckmann et al., 2013*; *Green et al., 2015*; *Land, 2015*), plants (*Karban and Shiojiri, 2009*; *Holopainen and Blande, 2012*; *Wilkins et al., 2014*), and in unicellular organisms including amoebae (*Hirose et al., 2011*; *Hirose et al., 2015*) and bacteria (*Marraffini and Sontheimer, 2010*; *Pathak et al., 2013*; *Le and Otto, 2015*). The precise molecular and structural basis underlying the evolution of kin discrimination, however, is largely unexplored. In social amoebae (*Hirose et al., 2011*), bacteria (*Pathak et al., 2013*; *Cao and Wall, 2017*), and yeast (*Kraushaar et al., 2015*), cell surface adhesins have been suggested to confer kin discrimination by mediating adhesion between cells using either homotypic or heterotypic interactions. Homotypically acting adhesins appear to be particularly suited for kin discrimination, because they depend on only one gene and therefore always select for interaction partners that carry the exact same gene. The

key question of how exactly cell surface adhesins can evolve at the structural level to permit diversification without compromising the ability to confer homotypic interaction has not yet been fully addressed.

Yeasts are ideal models to study the evolution of adhesin-based kin discrimination, because they are genetically tractable and harbor a family of cell surface adhesins that confer mating of sexual cells, the formation of social aggregates and the colonization of hosts (*Dranginis et al., 2007*; *Brückner and Mösch, 2012*; *de Groot et al., 2013*). Flo11-type adhesins are of interest in the context of kin discrimination, because Flo11 from the budding yeast *Saccharomyces cerevisiae* (ScFlo11) confers cell aggregation by homotypic protein-protein interaction (*Bayly et al., 2005*; *Goossens and Willaert, 2012*; *Kraushaar et al., 2015*; *Barua et al., 2016*; *Figure 1A*). Moreover, Flo11 confers the formation of diverse multicellular growth forms representing cooperative communities including biofilms, invasive filaments or adhesive aggregates (*Lo and Dranginis, 1998*; *Cullen and Sprague, 2000*; *Reynolds and Fink, 2001*; *Chow et al., 2019*). It has also been shown that this adhesin confers competitive advantages through cell differentiation in clonal biofilms and might lead to kin discrimination in mixed populations (*Regenberg et al., 2016*; *Oppler et al., 2019*). Importantly, the structure of the ScFlo11A adhesion domain has been solved at high resolution (*Kraushaar et al., 2015*), enabling comparative structural analysis of diverse and distantly related Flo11-type domains.

Here, we explored the possibility how the adhesion domain of Flo11 from *S. cerevisiae* and related Flo11-type proteins from other yeasts confer aggregation of cell populations and mediate kin discrimination at single cell level. For this purpose, we developed novel methods to (i) measure the adhesion forces between individual Flo11A-presenting yeast cells by atomic force microscope (AFM)-based single cell force spectroscopy (SCFS) (*Benoit et al., 2000*; *Krieg et al., 2008*), (ii) determine the efficiency of Flo11A-mediated cell aggregation using microscopy-based quantification of large numbers of cell aggregates, and (iii) quantify the Flo11A-dependent segregation of competing yeast strains in heterogenous biofilms. Our study shows that Flo11A domains from a variety of different yeast species confer efficient cell aggregation by facilitating homotypic interactions. The structural and mutational analysis of different Flo11A domains demonstrates that homotypic Flo11A-Flo11A interactions depend on evolutionary conserved aromatic residues, which form two bands at the Flo11A surface. Finally, we observe that heterotypic interactions between Flo11A domains from different yeast species or *S. cerevisiae* strains confer weak adhesion forces and efficient strain separation, suggesting that in social yeasts Flo11A-mediated adhesion is employed for kin discrimination at species and sub-species levels.

## Results

### The capacity of Flo11A domains for cellular adhesion is highly conserved in Saccharomycetales

We wanted to determine whether the capacity of Flo11-type proteins to confer cellular adhesion is not only found in *S. cerevisiae*, but also in other yeast species. In addition, we were interested in functional diversity of Flo11A domains within the species of *S. cerevisiae*. For this purpose, we constructed an evolutionary tree, which is based on 16 sequences from 11 different yeast species within the order of Saccharomycetales and on Flo11A sequences from 52 *S. cerevisiae* strains (*Figure 1B* and *Supplementary file 1*). The analysis revealed that Flo11A domains displayed a high degree of sequence variability (*Figure 1—figure supplement 1*). Moreover, *S. cerevisiae* Flo11A alleles were well separated from Flo11 variants of other Saccharomycetales with the exception of *Saccharomyces paradoxus*, a non-domesticated *Saccharomyces* species and closest relative of *S. cerevisiae* (*Liti et al., 2009*; *Boynton and Greig, 2014*). We also discovered that 19 of the 52 *S. cerevisiae* Flo11A alleles carry a particular small insert of about 15 amino acids, which is absent in Flo11-type sequences from other yeast species (*Figure 1B* and *Figure 1—figure supplement 1*). Remarkably, Flo11A variants carrying this extra insert were exclusively found in the 'Sake' lineage of non-mosaic *S. cerevisiae* strains (*Liti et al., 2009*; *Boynton and Greig, 2014*).

For functional analysis, we selected a total of 8 different Flo11A-type domains from *S. cerevisiae* and *S. paradoxus*, the industrially relevant yeasts *Kluyveromyces lactis*, *Torulospora delbrueckii*, and *Komagataella pastoris*, as well as from the human pathogenic yeasts *Clavispora lusitaniae* and

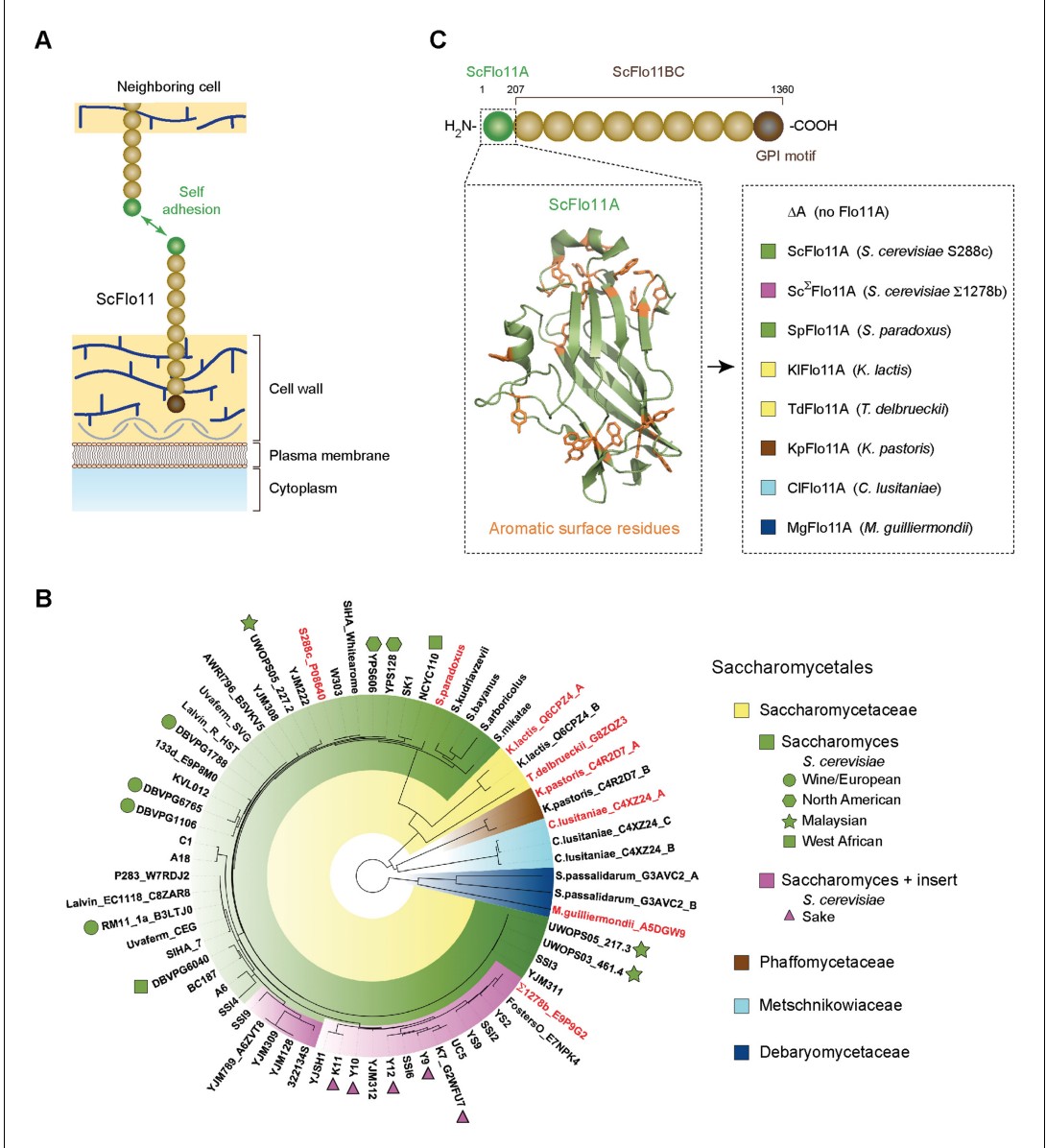

**Figure 1.** Functional model and phylogeny of fungal Flo11A domains. (**A**) Model showing self adhesion of two *S. cerevisiae* Flo11 molecules (ScFlo11) conferred by homotypic binding of ScFlo11A adhesion domains (green sphere), which are covalently anchored via ScFlo11BC (beige spheres) and a transamidated GPI motif (brown sphere) to the cell wall (***Kraushaar et al., 2015***). (**B**) Phylogenetic tree of Flo11A-type domains from different Saccharomycetales species. The tree was created by using the webPRANK alignment server (***Löytynoja and Goldman, 2010***) and 52 Flo11A sequences from different *S. cerevisiae* strains together with 16 Flo11A-type sequences from *S. paradoxus*, *S. kudriavzevii*, *S. bayanus*, *S. arboricolus*, *S. mikatae*, *K. lactis*, *T. delbrueckii*, *K. pastoris*, *C. lusitaniae*, *S. passalidarum* and *M. guilliermondii*, respectively. Affiliations of Flo11A-type sequences with the families of Saccharomycetaceae, Phaffomycetaceae, Metschnikowiaceae or Debaryomycetaceae are indicated by colors. Absence (green) or presence (magenta) of a small insert of about 15 amino acids in Flo11A sequences from Saccharomyces species is indicated. For Flo11A sequences from *S. cerevisiae*, names of corresponding strains are indicated followed by the UniProt database identification number, where available. *S. cerevisiae* strains with clean lineages, which according to whole genome analysis form five well-supported clades (Wine/European; North American; Malaysian; West African; Sake) (***Liti et al., 2009***), are indicated by different symbols. For all other Flo11A-type sequences, names of corresponding yeast species are shown followed by the UniProt identification number. For Flo11-type proteins carrying multiple Flo11A domains, position of the domain relative to the N-terminus is indicated (A: first; B: second; C: third). Flo11A-type domains functionally characterized in *S. cerevisiae* in this study are shown in red. (**C**) Diagram showing the architecture of ScFlo11 (top) together with the structure of the ScFlo11A adhesion domain with the aromatic surface residues (***Kraushaar et al., 2015***) (left inlet) and seven Flo11A domains from different Saccharomycetales species that were replaced for ScFlo11A (right inlet, colored according to **B**). Numbers above diagram indicates amino acid residues flanking the A and BC region of ScFlo11. The length of the A region of different species varies between 185 and 240 amino acids (***Figure 1—figure supplement 1***).

The online version of this article includes the following figure supplement(s) for figure 1:

*Figure 1 continued on next page*

*Figure 1 continued*

**Figure supplement 1.** Multiple sequence alignment of Flo11A-type domains from different Saccharomycetales species.
**Figure supplement 2.** Functional analysis of Flo11A-type domains from different Saccharomycetales species.

*Meyerozyma guilliermondii*. These Flo11A domains were introduced into an *S. cerevisiae* expression system (*Figure 1C* and *Supplementary files 2*, *3*), which is based on *S. cerevisiae* ScFlo11 and allows presentation of adhesion domains at the cell surface for functional analysis (*Veelders et al., 2010*; *Kraushaar et al., 2015*). Remarkably, all Flo11A domains tested were competent to confer adhesive growth, even though their protein sequences are highly variable (*Figure 1—figure supplement 1* and *Figure 1—figure supplement 2*). Because, their degree of identity to ScFlo11A can be as low as 18%, our finding indicates that the capacity of Flo11A domains for conferring cellular adhesion is highly conserved in Saccharomycetales.

## Homotypic ScFlo11A interactions confer strong intercellular adhesion and efficient cell-cell aggregation

Standard adhesive growth assays are unable to distinguish between cell-cell and cell-surface adhesion and do not provide quantitative information at the single cell level. We therefore employed two methods to quantify (i) the adhesion forces between individual Flo11A-presenting yeast cells and (ii) the efficiency of Flo11A-mediated cell aggregation in populations. We first analyzed ScFlo11A, because it confers cell-cell aggregation by homotypic protein-protein interactions (*Kraushaar et al., 2015*). For measuring adhesion forces between two yeast cells, we advanced existing SCFS protocols (*Figure 2*, *Figure 2—figure supplement 1* and *Figure 2—figure supplement 2*). Specifically, single yeast cells presenting ScFlo11A were attached on a cantilever (probe cell) and on a glass surface (target cell). The adhesion forces were determined by bringing both cells into contact and applying a force of 1 nN for varying time periods followed by the retraction of the probe from the target cell until both cells separated abruptly (*Figure 2A–D*). We found an average maximal adhesion force ($F_{max}$) between both ScFlo11A-presenting cells of 13.4 nN, whereas the adhesion forces between ScFlo11A-presenting and non-presenting cells ranged significantly lower by almost an order of magnitude (*Figure 2D*). Thus, the strength of cell-cell adhesion force is high if both cells carry the ScFlo11A domain. Kinetic cell-cell adhesion measurements using increasing contact times ranging from 0.5 s to 20 s confirm this finding and reveal that adhesion forces between Flo11-presenting cells are established within the first second of cell-cell contact and steadily increase up to 20 s contact time (*Figure 2—figure supplement 3*). $F_{max}$ values produced by contact times exceeding 20 s could not be reliably assayed, because the probe cells were repeatedly torn from the cantilever. This indicates that the adhesion forces established by homophilic ScFlo11A-ScFlo11A interactions considerably exceed 13.4 nN.

To determine the correlation between ScFlo11-mediated adhesion forces at the single cell level and the degree of cell-cell aggregation in homogeneous populations(groups of genetically identical yeast cells), we introduced quantitative cell aggregate microscopy (QCAM) (*Figure 2—figure supplement 4*). QCAM analysis shows that ScFlo11A-presenting cells are able to form large aggregates covering areas of up to 1 mm$^2$ (*Figure 2E*; aggregate class IV). In contrast, areas covered by aggregates formed from cells lacking ScFlo11A were more than 1000-fold smaller (*Figure 2E*; aggregate class I). The data demonstrates a clear correlation between adhesion forces mediated by ScFlo11A at the single cell level and cell-cell aggregation in populations.

## Two conserved aromatic bands on Flo11A surface contribute to cell-cell adhesion

The ScFlo11A structure contains sixteen conserved aromatic residues at the protein surface, which are arranged in two bands (aromatic bands I and II) (*Figure 3A* and *Figure 1—figure supplement 1*; *Kraushaar et al., 2015*). We determined their functional requirement by systematic exchange for aspartates in seven different mutational groups. This analysis identified a total of twelve aromatic surface residues involved in adhesive growth (*Figure 3—figure supplement 1*). To discriminate between the requirement of aromatic side chains and the presence of negative charges introduced by aspartates, we further created variants of ScFlo11A carrying alanines instead of aromatic residues.

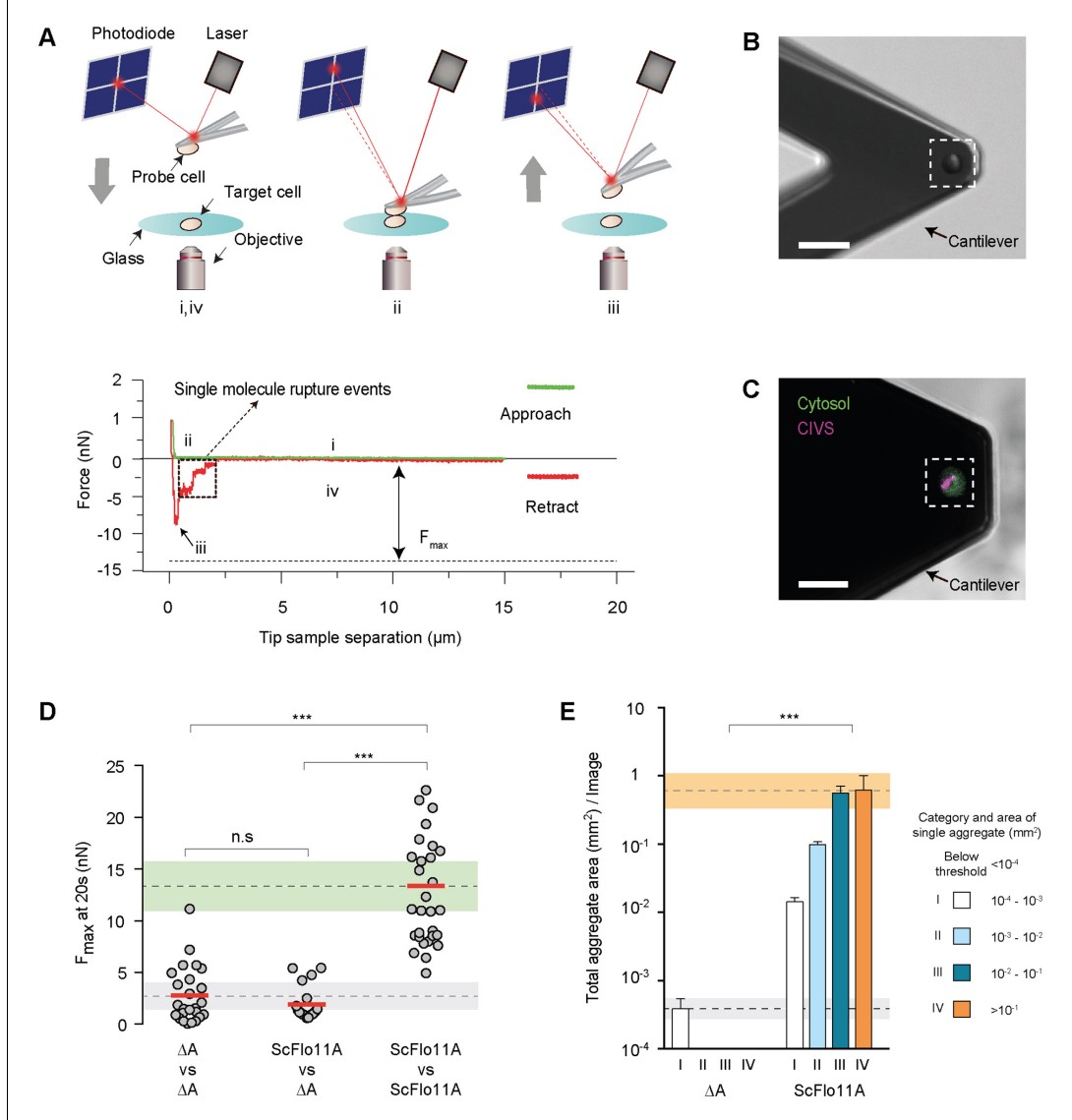

**Figure 2.** Quantification of Flo11-mediated cell adhesion forces. (A) Outline of a single-cell force spectroscopy (SCFS) experiment (upper scheme). (i) The probe cell is immobilized on an AFM cantilever and brought into contact at a defined speed with a target cell adhering to a glass substrate until a preset contact force is reached. After a defined contact time (ii), the cantilever is retracted until the probe cell is fully separated from the target cell (iii and iv). During approach and retraction, the cantilever deflection and thus, the force acting on the probe cell, is detected by using a laser and a photodiode and is recorded in a force-distance curve (lower scheme) that allows calculation of the maximum adhesion force ($F_{max}$). (B) Single yeast probe cells immobilized on an AFM cantilever. The image was obtained by differential interference contrast imaging. Scale bar corresponds to 15 μm. (C) Confocal laser scan microscopy after staining with FUN-1 dye. Fungal cells internalize FUN-1 and the dye is seen as diffuse green cytosolic fluorescence. FUN-1 is then transported to the vacuole in metabolically active cells and subsequently compacted into fluorescent, red cylindrical intravacuolar structures (CIVS), here pseudo-colored in purple to indicate healthy cells (*Millard et al., 1997*). Scale bar corresponds to 15 μm (D) Adhesion forces mediated by ScFlo11A at single cell level were determined for homotypic or heterotypic interaction of yeast cells presenting no Flo11A (ΔA) or ScFlo11A by SCFS (*Figure 2—source data 1*). Average $F_{Max}$ values measured after 20 s of contact time are shown as red bars and were calculated from at least 15 independent individual measurements (grey dots). The average adhesion forces mediated by cells lacking ScFlo11A or with ScFlo11A are shown by dotted lines, and corresponding SD areas are shown by grey and green bands. (E) Cell-cell aggregation strength mediated by ScFlo11A in homogeneous populations was determined using yeast strains presenting no Flo11A (ΔA) or ScFlo11A by QCAM (*Figure 2—figure supplement 4*). Total area covered (mm²) by all cell aggregates of a given size category (I - IV) per image area (1 cm²) is shown as a quantitative measure for cell-cell aggregation. Error bars indicate standard deviation obtained by at least three independent measurements. The average total aggregate areas obtained with cells lacking Flo11A or with regular ScFlo11A are shown by dotted lines, and corresponding SD areas are shown by grey and orange bands. Significance was calculated applying an unpaired t-test (D) or a Wilcoxon rank sum test (E) with p>0.05 (n.s), 0.05 ≥ P > 0.01 (*), 0.01 ≥ P > 0.001 (**), p≤0.001 (***).

*Figure 2 continued on next page*

*Figure 2 continued*

The online version of this article includes the following source data and figure supplement(s) for figure 2:

**Source data 1.** Single cell-cell adhesion forces determined by SCFS and presented in *Figure 2*.
**Figure supplement 1.** Preparation of single yeast cells suitable for SCFS analysis.
**Figure supplement 2.** Controls for cell adhesion on cantilever.
**Figure supplement 3.** Time course of average force values obtained by single-cell force spectroscopy (SCFS).
**Figure supplement 4.** Quantitative cell aggregation microscopy (QCAM).

We found no significant impact on adhesive growth upon substituting groups of two to three aromatic residues of either band I or band II. However, concomitant mutations in bands I and II (Y111A Y113A Y118A Y133A W144A Y196A) led to a significant decrease of cellular adhesion.

To determine the requirement of aromatic residues in ScFlo11A for cell-cell adhesion at single cell level and in homogeneous populations, we performed SCFS and QCAM with six selected variants. We found that mutations to aspartate outside the aromatic bands (W51D Y182D) have no effect. In contrast, aspartate mutations within band I (Y111D Y113D Y118D) or band II (Y133D W144D Y196D) alone significantly reduced homotypic and heterotypic cell-cell interactions as well as cell-cell adhesion in populations (*Figure 3B,C*, *Figure 2—figure supplement 3* and *Figure 3—figure supplement 1*). Analogous mutations to alanine in either band I or band II showed a clear reduction at single cell level in homotypic configuration, but had only minor effects at population level. Finally, a combination of alanine mutations in both bands I and II (Y111A Y113A Y118A Y133A W144A Y196A) abrogate homotypic single cell-cell adhesion as well as cell aggregate size. Similarly, cell-cell adhesion was completely abolished when testing alanine mutations in band I against alanine mutation in band II in a heterotypic configuration. Taken together, the data indicate that the two conserved aromatic bands at the ScFlo11A surface contribute to cell-cell adhesion at the single cell level as well as in populations and suggest that they directly interact.

## Flo11A from different yeast species confer kin discrimination at species level

Given our finding that Flo11A-mediated cellular adhesion appears to be well-conserved in Saccharomycetales, we wondered to which extent structures are conserved. Therefore, the crystal structure of Flo11A from *K. pastoris* (KpFlo11A), a variant from the family of Phaffomycetaceae, was determined at a resolution of 1.40 Å (*Figure 4—figure supplement 1*, *Supplementary file 4* and *Supplementary file 5*). Despite a sequence identity between KpFlo11A and ScFlo11A of only 32%, their overall structures showed a high degree of similarity with an r.m.s.d. of 1.01 Å for 102 Cα atoms after superposition (*Figure 4A*, *Figure 4—figure supplement 1* and *Figure 4—figure supplement 2*). The core motif of KpFlo11A, the FNIII domain, showed the highest degree of similarity (0.70 Å for 61 Cα atoms derived from the β-sheet), whereas the neck-like subdomain and the apical regions deviated from ScFlo11A. The surface of KpFlo11A exposed 6 tryptophans and nine tyrosines that cluster in two bands, which were similar but not identical to the aromatic bands of ScFlo11A (*Figure 4A* and *Figure 4—figure supplement 2*). Also, both the KpFlo11A and ScFlo11A surfaces were quite rigid given the low thermal B factors found for their surface-exposed aromatic residues (*Figure 4—figure supplement 3*). Finally, a ConSurf analysis (*Celniker et al., 2013*) of different Saccharomycetales Flo11A domains showed that the FNIII domain is well conserved, even when sequences below 50% sequence identities were used, whereas the apical regions were less well conserved (*Figure 1—figure supplement 1* and *Figure 4—figure supplement 3*). Thus, the core and aromatic surface patterns of even highly divergent Flo11A domains appeared to be conserved, explaining the functional conservation with respect to cellular adhesion.

For ScFlo11A, we previously found homophilic in vitro binding with a $K_D$ of 19.5 μM (*Kraushaar et al., 2015*). To test, whether KpFlo11A can also establish homophilic interactions, we performed SPR analysis. Similar to ScFlo11A, KpFlo11A is able to undergo homophilic interactions with a $K_D$ of 31.8 μM (*Figure 4B* and *Figure 4—figure supplement 4*). We also measured possible heterophilic interactions between ScFlo11A and KpFlo11A, where we found a clearly decreased interaction with a $K_D$ of 121.1 μM (*Figure 4B* and *Figure 4—figure supplement 4*). A further kinetic analysis showed comparable dissociation constants ($k_{off}$) for both homo- and heterophilic binding. In

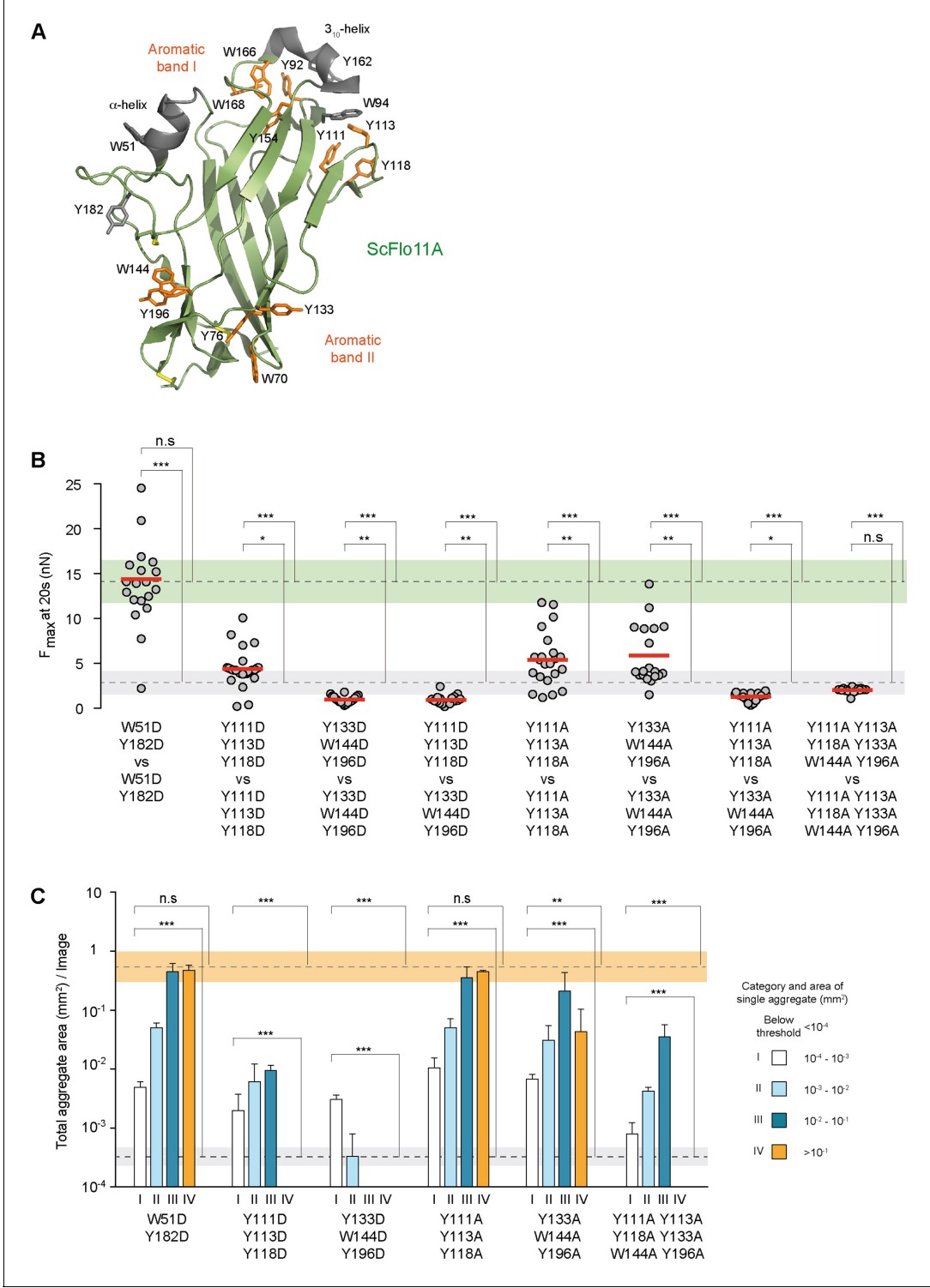

**Figure 3.** Functional mapping of the Flo11A protein surface at single cell and population level. (**A**) Structural model of ScFlo11A (PDB code 4UYR). Functionally relevant aromatic surface residues of aromatic bands I and II are depicted in orange (as shown in *Figure 3—figure supplement 1*). Surface exposed aromatic residues and helices not required for cell-cell adhesion, are shown in grey. Disulfide bonds are shown in yellow. (**B**) Adhesion forces mediated by ScFlo11A mutants at single cell level were determined by SCFS (*Figure 3—source data 1*) as described in *Figure 2D*. Mutations of ScFlo11A are indicated. The average adhesion forces and SD areas mediated by cells with regular ScFlo11A or lacking ScFlo11A are indicated as green or grey bands. (**C**) Cell-cell aggregation strength mediated by ScFlo11A mutants in homogeneous populations was determined by QCAM as described *Figure 3 continued on next page*

*Figure 3 continued*

in *Figure 2E*. Mutations of ScFlo11A are indicated and correspond to mutations measured by SCFS in b. The average total aggregate areas obtained by cells with regular ScFlo11A or lacking ScFlo11A are indicated as orange or grey bands. Significance was calculated as described in *Figure 2*.

The online version of this article includes the following source data and figure supplement(s) for figure 3:

**Source data 1.** Single cell-cell adhesion forces determined by SCFS and presented in *Figure 3*.

**Figure supplement 1.** Structure-based functional analysis of ScFlo11A.

contrast, the association constant ($k_{on}$) for heterophilic binding was significantly lower compared to homophilic binding. Thus, both ScFlo11A and KpFlo11A are competent for efficient homophilic interactions in vitro, but they are able to clearly discriminate between each other.

To determine the in vivo function of KpFlo11A during adhesion at the single cell level and in homogeneous populations, we performed SCFS and QCAM using an *S. cerevisiae* expression system (see above). We found that KpFlo11A confers efficient single cell-cell interactions with $F_{max}$ values that were comparable to those of ScFlo11A (*Figure 4C* and *Figure 2—figure supplement 3*). Similarly, KpFlo11A enabled efficient cell aggregation at the population level (*Figure 4D*). We next tested heterotypic cell-cell interactions using KpFlo11A and ScFlo11A, showing clearly reduced $F_{max}$ values in comparison to the homotypic configurations (*Figure 4C* and *Figure 2—figure supplement 3*). Finally, we measured homotypic and heterotypic interactions using Flo11A from *C. lusitaniae* (ClFlo11A), a further variant from the family of Metschnikowiaceae that shows only 29% identity to ScFlo11A and 38% identity to KpFlo11A. This analysis showed that in the homotypic configuration ClFlo11A confers a cell-cell adhesion comparable to the other two variants, but clearly discriminates in heterotypic situations (*Figure 4C,D* and *Figure 2—figure supplement 3*).

In order to test, whether Flo11A is able to confer allele-specific discrimination in heterogeneous populations (groups of genetically diverse cells), we performed competitive biofilm assays with yeast strains expressing no Flo11A, ScFlo11A, KpFlo11A or ClFlo11A, respectively, and different fluorescent markers (GFP or RFP) for genotypic tracking, in all possible combinations (*Figure 5*). Here, we found significant differences between homotypic (single Flo11A variant) and heterotypic (two different Flo11A variants) biofilms with respect to the ratio of the Flo11A expressing strains at the outer edge. In all homotypic biofilms, this ratio was found to be very low (*Figure 5B*), and the sectoring observed indicates genetic drift (*Hallatschek et al., 2007*). This demonstrates that strains expressing the same Flo11A variant and differing only in the fluorescent marker are both present in significant amounts at the outer biofilm edge and do not efficiently outcompete each other. In contrast, the ratio between strains expressing different Flo11A variants (heterotypic biofilms) was found to be significantly higher showing that one allele outcompetes the other and monopolizes the outer edge of the expanding biofilm. This indicates that heterogenous Flo11A alleles are able to discriminate against each other in heterogeneous populations. This conclusion is supported by the finding that in all heterotypic biofilms one of the alleles clearly dominates the biofilm with respect to its relative presence within the whole area (*Figure 5—figure supplement 1A,C*). In addition, the total sizes of mixed biofilms generally correspond to the sizes of the homogeneous biofilms of dominant alleles (*Figure 5—figure supplement 1A,B*).

In summary, the data of this section suggest that Flo11A domains from different yeast species with similar overall structures confer kin discrimination at single cell level and in heterogenous populations.

## Flo11A from different *S. cerevisiae* strains confer kin discrimination at sub-species level

Our sequence analysis reveals that in *S. cerevisiae* a significant number of Flo11A variants carry a unique insert of about 15 amino acids. We therefore created a 3D model of one of these variants from the strain $\Sigma$1278b (Sc$^\Sigma$Flo11A) (*Dowell et al., 2010*). This model showed that the additional insert ($\Sigma$-insert) is located at the protein surface in vicinity of aromatic band I, which is conserved in Sc$^\Sigma$Flo11A together with aromatic band II (*Figure 6A*).

To analyze the consequences of the $\Sigma$-insert, we measured in vivo functionality of Sc$^\Sigma$Flo11A that differs from ScFlo11A by the 15 amino acid $\Sigma$-insert and an additional 11 single amino acid exchanges (*Figure 1—figure supplement 1*) This analysis revealed that the Sc$^\Sigma$Flo11A variant shows

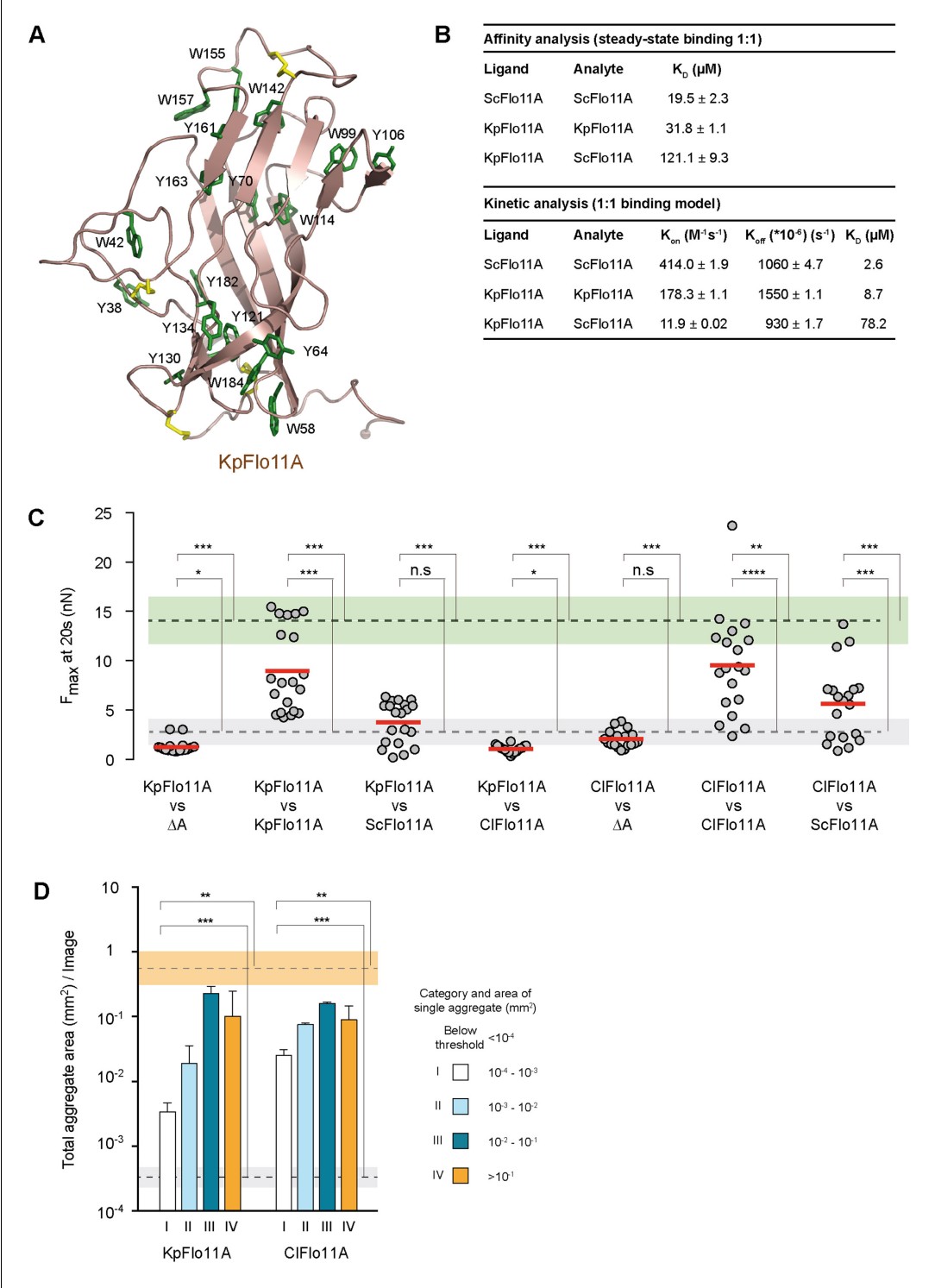

**Figure 4.** Structural and functional properties of Flo11A from *Komagataella pastoris* at single cell and population level. (A) Crystal structure of KpFlo11A at 1.40 Å resolution (PDB code 5FV5). The structural model shows solvent-exposed aromatic residues (Tyr, Trp) in green. Disulfide bonds are shown in yellow. (B) Results and statistics derived from SPR analysis (*Figure 4—figure supplement 4*) for ScFlo11A-ScFlo11A, KpFlo11A-KpFlo11A and KpFlo11A-ScFlo11A interactions, respectively. Calculations for affinity and kinetic analysis are based on experiments carried out at least in duplicate. The $K_D$ value for ScFlo11A in the affinity analysis was previously reported (*Kraushaar et al., 2015*). (C) SCFS analysis for measuring adhesion forces mediated by homo- or heterotypic interactions of KpFlo11A, ScFlo11A and ClFlo11A (*Figure 4—source data 1*), respectively, was performed as described in *Figure 2D*. (D) QCAM analysis for measurement of cell-cell aggregation strength mediated by KpFlo11A and ClFlo11A, respectively, was

*Figure 4 continued on next page*

*Figure 4 continued*

performed as described in *Figure 2E*. The average total aggregate areas obtained by cells with regular ScFlo11A or lacking ScFlo11A are indicated as orange or grey bands. Significance was calculated as described in *Figure 2*.

The online version of this article includes the following source data and figure supplement(s) for figure 4:

**Source data 1.** Single cell-cell adhesion forces determined by SCFS and presented in *Figure 4*.
**Figure supplement 1.** Overall structure of the KpFlo11A domain from *Komagataella pastoris*.
**Figure supplement 2.** Structural comparison between ScFlo11A and KpFlo11A.
**Figure supplement 3.** Rigidity of aromatic surface patches in ScFlo11A and KpFlo11A and overall conservation.
**Figure supplement 4.** SPR binding analysis of KpFlo11A-KpFlo11A and KpFlo11A-ScFlo11A interactions.

enhanced adhesive growth formation, confers significantly higher $F_{max}$ in homotypic cell-cell adhesion and leads to stronger cell aggregation (*Figure 6B,C* and *Figure 6—figure supplement 1*). We further measured heterotypic interactions of single cells carrying $Sc^\Sigma Flo11A$ in combination with cells carrying ScFlo11A, revealing that these two Flo11A variants were unable to produce substantial adhesion forces (*Figure 6B*). Similarly, $Sc^\Sigma Flo11A$ was unable to interact with the Flo11A variants from other species, KpFlo11A and ClFlo11A (*Figure 2—figure supplement 3*). We also tested the role of the $\Sigma$-insert in mediating sub-species kin discrimination. For this purpose, we introduced the $\Sigma$-insert into ScFlo11A creating the hybrid variant $ScFlo11A^{\Sigma ins}$. Functional analysis showed that this synthetic variant, which still differs from regular $Sc^\Sigma Flo11A$ by 11 single amino acid exchanges, is fully competent to mediate adhesive growth, homotypic cell-cell interaction and cell aggregation (*Figure 6B,C* and *Figure 6—figure supplement 1*). More importantly, heterotypic single cell analysis showed that $ScFlo11A^{\Sigma ins}$ significantly interacts with $Sc^\Sigma Flo11A$, but not with ScFlo11A, demonstrating that the transfer of the $\Sigma$-insert is sufficient to transfer allele-specific discrimination (*Figure 6B* and *Figure 2—figure supplement 3*). Finally, we also performed competitive biofilm assays with strains expressing $Sc^\Sigma Flo11A$ (*Figure 5* and *Figure 5—figure supplement 1*). This analysis clearly shows that this allele outcompetes all other Flo11A variants tested with respect to its presence at the outer biofilm edge and within the whole biofilm. Together, these findings clearly support the view that Flo11A is able to confer allele-specific discrimination in heterogeneous populations at sub-species level.

## Discussion

Here, we present evidence that in social yeasts Flo11-type adhesins confer kin discrimination at species and sub-species level. By employing biophysical (SCFS), cell aggregation (QCAM) measurements and competitive biofilm assays, we demonstrate that Flo11-type domains from diverse yeasts confer efficient cell-cell adhesion at single cell level and in populations in homotypic configurations and are able to efficiently discriminate between kin and non-kin variants in heterotypic situations in vivo. Our high-resolution structural analysis shows that the core and aromatic surface patterns of highly divergent Flo11A domains are similar, and our in vitro functional analysis demonstrates that different Flo11A variants discriminate between homophilic and heterophilic interactions. *FLO11*-type domains thus fulfill important criteria for single greenbeard genes, which confer benefits by homophilic interactions to cells or organisms that carry this allele (*Hamilton, 1964*; *Dawkins, 1976*).

To our knowledge, the Flo11A-type domain represents the first example for a fungal adhesin, which is able to mediate kin discrimination by homophilic interaction at both species and sub-species level. In *S. cerevisiae*, another type of cell surface adhesion domain, the PA14/Flo5 domain, has been demonstrated to confer kin discrimination (*Smukalla et al., 2008*). However, this type of domain confers cell-cell adhesion by heterotypic interaction with mannan oligosaccharide chains present on the surface of neighboring cells (*Veelders et al., 2010*). Thus, the PA14/Flo5 domain mediates kin discrimination more indirectly, because two-way bonding between two cells bearing the domain is stronger than one-way bonding between a cell exposing the domain and a cell that lacks it. This mode of discrimination is different from homophilic interactions between identical Flo11A domains, which must be present on the surface of both interacting cells.

In another social microbe, the cell adhesion gene *csaA* of *Dictyostelium* has been discussed to confer greenbeard recognition by homotypic interaction between bearing cells. However, *csaA* is not considered to confer kin discrimination in nature due to the lack of significant variability across

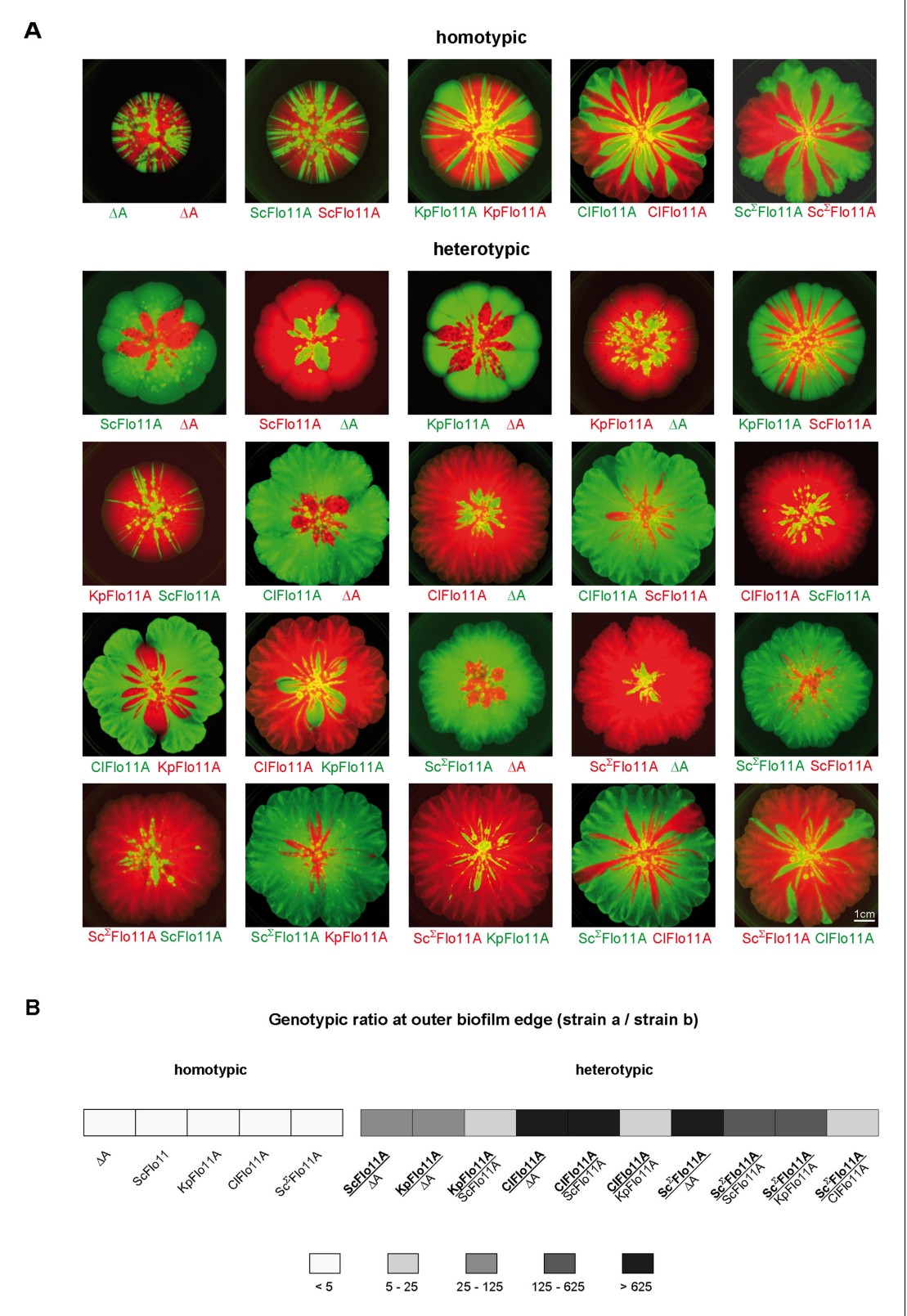

**Figure 5.** Flo11A-mediated separation of competitive yeast strains. (**A**) Competitive biofilm assays. Yeast strains expressing the indicated Flo11 variants were mixed in a 1:1 ratio and were grown to a mat biofilm for three weeks. To distinguish between competing Flo11 variants, strains were tagged by expression of either GFP (green color) or RFP (red color). Homotypic and heterotypic Flo11 combinations are indicated. Each Flo11 variant was assayed using two independent strains tagged by either GFP or RFP, to test robustness to the fluorescence marker. Scale bar corresponds to 1 cm. (**B**)
*Figure 5 continued on next page*

*Figure 5 continued*

Quantification of presence of competing Flo11 strains from *A* present at the outer biofilm edge (*Figure 5—source data 1*). For quantification of the genotypic ratio, the amount of the signal produced by the superior strain a (bold) was divided by the amount of the signal from the inferior strain b (regular). Average ratios are presented using five different categories as indicated. For each combination of competing strains, at least two biological replicates of independently obtained clones were assayed.

The online version of this article includes the following source data and figure supplement(s) for figure 5:

**Source data 1.** Quantification of RFP and GFP signals in mixed biofilms presented in *Figure 5*.

**Figure supplement 1.** Quantitative analysis of biofilms upon competitive assays.

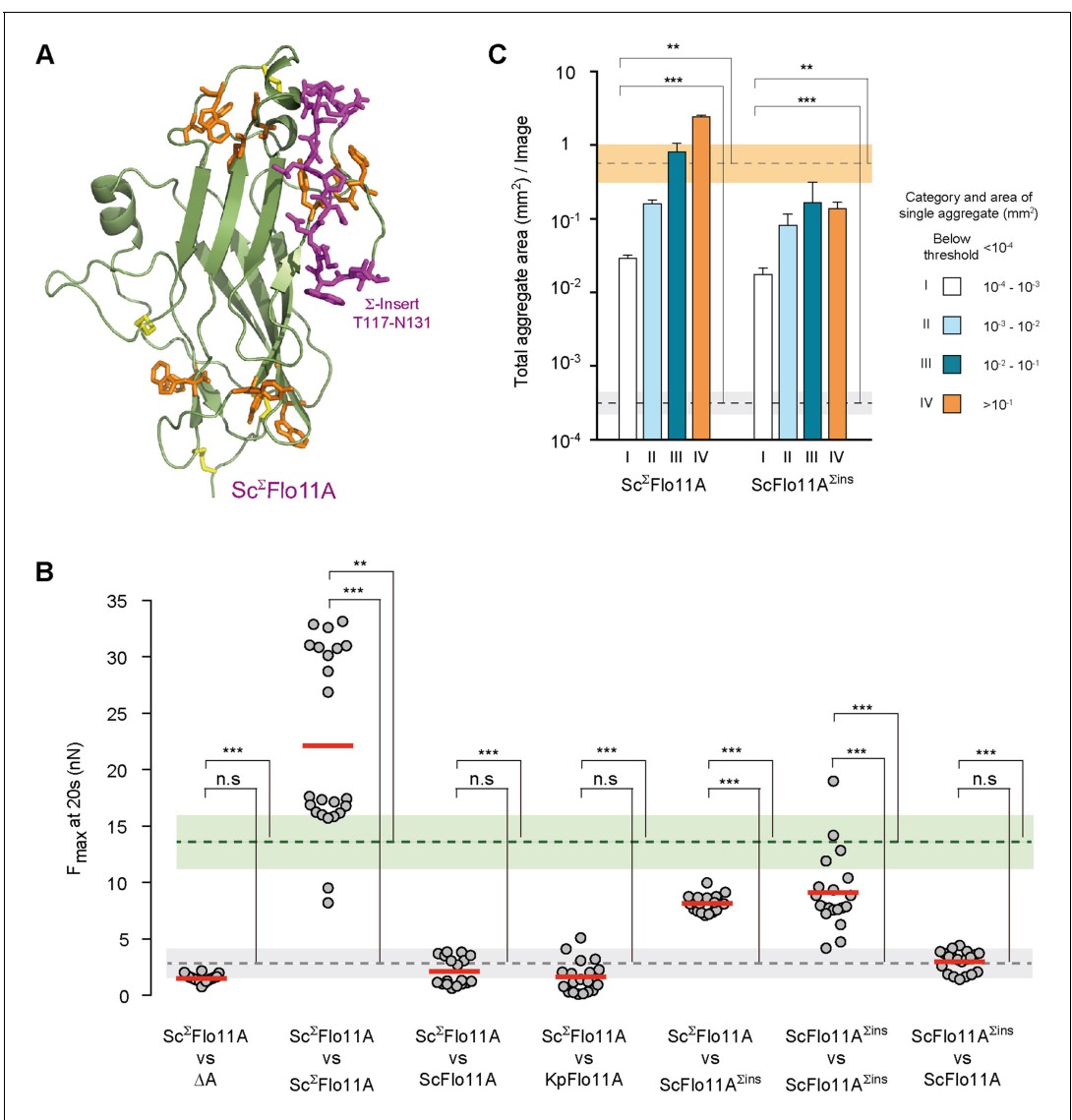

**Figure 6.** Structural and functional properties of Sc$^\Sigma$Flo11A from S. cerevisiae strain $\Sigma$1278b at single cell and population level. (**A**) A structural model of Sc$^\Sigma$Flo11A as obtained as snap shot by molecular dynamics analysis. The Sc$^\Sigma$Flo11A-specific insert differing from ScFlo11A is shown in magenta. Aromatic surface residues functionally relevant in ScFlo11A (*Figure 3—figure supplement 1*), are shown in orange. Disulfide bonds are shown in yellow. (**B**) SCFS analysis for Sc$^\Sigma$Flo11A, ScFlo11A variants and KpFlo11A (*Figure 6—source data 1*). (**C**) QCAM analysis for Sc$^\Sigma$Flo11A and ScFlo11A variants.

The online version of this article includes the following source data and figure supplement(s) for figure 6:

**Source data 1.** Single cell-cell adhesion forces determined by SCFS and presented in *Figure 6*.

**Figure supplement 1.** Functional analysis of Sc$^\Sigma$Flo11A.

natural clones (*Queller et al., 2003*). More recent studies suggest that in social amoebae kin discrimination is conferred by the polymorphic proteins TgrB1 and TgrC1, which are thought to function as a heterotypic ligand-receptor pair (*Hirose et al., 2011*). In social bacteria, the TraA protein of *Myxococcus xanthus* is a polymorphic cell surface adhesin that confers kin discrimination by homotypic interaction (*Cao and Wall, 2017*). In contrast to our study, however, neither biophysical single cell analysis nor high-resolution structure-function relationships have been performed with these microbial kin discriminatory molecules.

A central outcome of our SCFS-based study are quantitative in vivo measurements of adhesion between two cells expressing fungal adhesins of both heterophilic and homophilic pairs. The Flo11A domain in certain yeast species confers adhesion forces of >25 nN between two cells after only 20 s contact time. This adhesion force is considerably higher than the adhesion forces measured between mammalian cells at similar contact times (*Krieg et al., 2008*; *Beckmann et al., 2013*; *Kragl et al., 2016*). Our quantitative SCFS assay in combination with structural and genetic approaches provides novel insights into how individual amino acids of the Flo11A domains contribute to cell−cell recognition and to the formation and strengthening of cell−cell adhesion at the single cell level. This approach provided previously unseen mechanistic insights at the single cell level of how Flo11A domains contribute to partner selection and discrimination through differential cell−cell adhesion over time. Furthermore, the quantitative SCFS data correlates with the phenotypes observed in homogeneous populations as analyzed by QCAM (*Figure 7*). Therefore, adhesion forces between cells determine macroscopic aggregate sizes.

How can Flo11-type adhesin mediated kin discrimination be visualized at the molecular level? In our previous work with ScFlo11A, we presented a model suggesting that Flo11A domains confer cell-cell adhesion by hydrophobin-like interactions via aromatic surface residues (*Kraushaar et al., 2015*). Our current study with several diverse Flo11A variants supports and further refines this model. First, the systematic mutational analysis of ScFlo11A clearly emphasizes the crucial importance of conserved aromatic residues for homotypic interactions and suggests their general

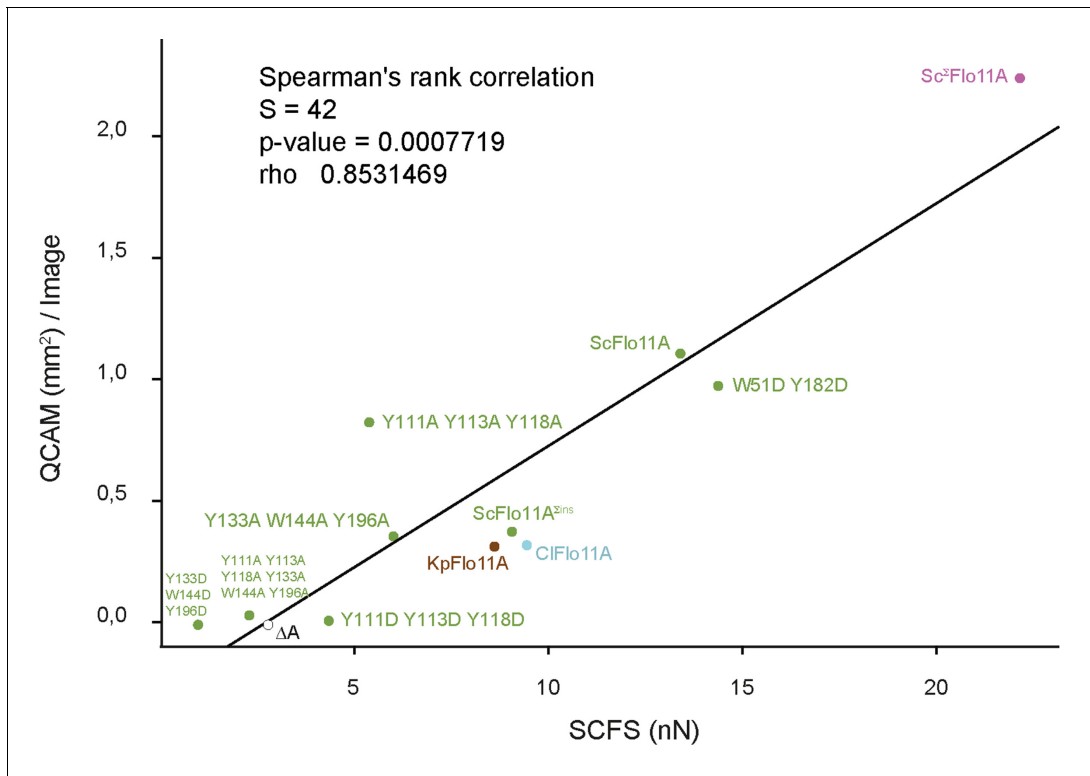

**Figure 7.** Correlation between SCFS and QCAM data. Correlation between maximum adhesion forces obtained by SCFS analysis and total aggregate areas obtained by QCAM analysis was calculated by Spearman's rank correlation test. Dots show SCFS and QCAM data obtained for different Flo11A variants as indicated.

functional relevance in other Flo11-type domains. Secondly, our study strongly suggests that in contrast to unspecific hydrophobins, Flo11A domains interact with each other in a much more specific manner. This conclusion is based on the significant differences that we observe when comparing (i) homotypic with heterotypic single cell-cell interactions and (ii) homophilic with heterophilic protein-protein interactions. Our comparative structural analysis further indicates that the precise topology of the aromatic surface residues might be an important feature for molecular discrimination. This notion is based on the finding that the ScFlo11A and KpFlo11A surface structures are highly divergent with the exception of multiple aromatic residues, which are arranged in a similar but not identical configuration. Moreover, even variants with nearly identical aromatic surface topologies, such as ScFlo11A and Sc$^\Sigma$Flo11A, can clearly discriminate between each other, given that a further surface feature is added, as exemplified by the $\Sigma$-specific insert.

An important question concerning the mechanism, by which homophilically interacting proteins are able to confer discrimination, addresses the contribution of molecular *cis*- and *trans*-interactions. In mammalian tissues and organs, teneurins, cadherins and nectins are well-studied adhesion molecules that confer homophilic and heterophilic cell-cell interactions by *cis*-clustering and *trans*-interactions (*Rikitake et al., 2012*; *Beckmann et al., 2013*; *Gul et al., 2017*). For protocadherins, a specialized family of cadherins, structural and functional as well as biophysical investigations have recently shown that these multi-domain cell-surface proteins confer neuronal self-recognition involving separate domains for *cis*- and *trans*-interactions (*Rubinstein et al., 2015*). Whether Flo11-mediated kin discrimination involves both *cis*- and *trans*-interacting molecules cannot be unambiguously answered yet. However, previous data suggests that Flo11-mediated cell-cell adhesion involves *trans*-interactions of collectives of Flo11 molecules formed by *cis*-clustering, rather than multiple bimolecular *trans*-interactions of Flo11 adhesins that do not interact in cis (*Figure 8*). Here, our previous ultrastructural analysis of Flo11-Flo11 contact sites support the view that Flo11 proteins form co-aligned fibers at the cell surface by clustering of Flo11 molecules in cis (*Kraushaar et al., 2015*). Moreover, the formation of such clusters could be mediated by *cis*-interaction of Flo11B domains via beta-aggregation-prone amyloid-forming sequences (*Ramsook et al., 2010*). Our SCFS analysis with mutated Flo11 variants also supports the view that tethering of Flo11 into collectives by *cis*-interactions might precede *trans*-interactions. Here, we found that Flo11 variants that carry mutations in either aromatic band I (I) or band II (II) are only partially abrogated for single cell-cell adhesion in homotypic configurations, whereas a complete loss of adhesion is observed in heterotypic measurements. This finding supports a model, in which productive adhesive forces can be achieved by either I-I or II-II *trans*-interactions, but they do not support the occurrence of I-II interactions. This further implies that Flo11 molecules must be tethered in cis before either I-I or II-II *trans*-interactions take place, given that a free choice between I-I or II-II would be assumed in the case of only bimolecular *trans*-interactions, which is not in agreement with the complete loss of adhesive forces observed in the heterotypic situation. However, because our localization of Flo11 on the cell surface appears to be evenly distributed, other models for Flo11-Flo11 *cis*- and/or *trans*-interactions are conceivable and need to be tested in the future.

How does cell surface presentation and interaction of Flo11-type proteins translate into the development of multicellular growth forms? A number of studies using diverse microbial systems have addressed the question of how cellular interactions affect the spatial structure, cooperation and competition in biofilms (*Nadell et al., 2016*). Our analysis of biofilm development indicates that in contrast to the formation of multicellular aggregates, the efficiency of biofilm formation as measured by the total size does not necessarily correlate with cell-cell adhesion forces. This is exemplified by comparing the properties of ScFlo11 and ClFlo11 adhesion domains in non-competitive situations. Here, ScFlo11A confers roughly 1.4-fold higher cell-cell adhesion forces, but on average leads to 1.8-fold lower biofilm sizes, when compared to ClFlo11A. This apparent discrepancy could be explained by postulating significant differences in cell-substrate adhesion conferred by these Flo11A variants. In other words, ClFlo11A might be able to interact with the substrate much more efficiently than ScFlo11A and thereby confer a significantly better spreading of the growing biofilm. It remains to be elucidated, whether and how different Flo11A variants confer adhesion to different substrate surfaces. Here, our study has provided sophisticated techniques, which might turn out to be useful for detailed analysis of cell-substrate adhesion forces at single cell level. Our study also allows to correlate cell-cell adhesion forces conferred by Flo11A domains with their ability to confer dominance in competitive situations such as mixed biofilms. Previous theoretical and experimental work has

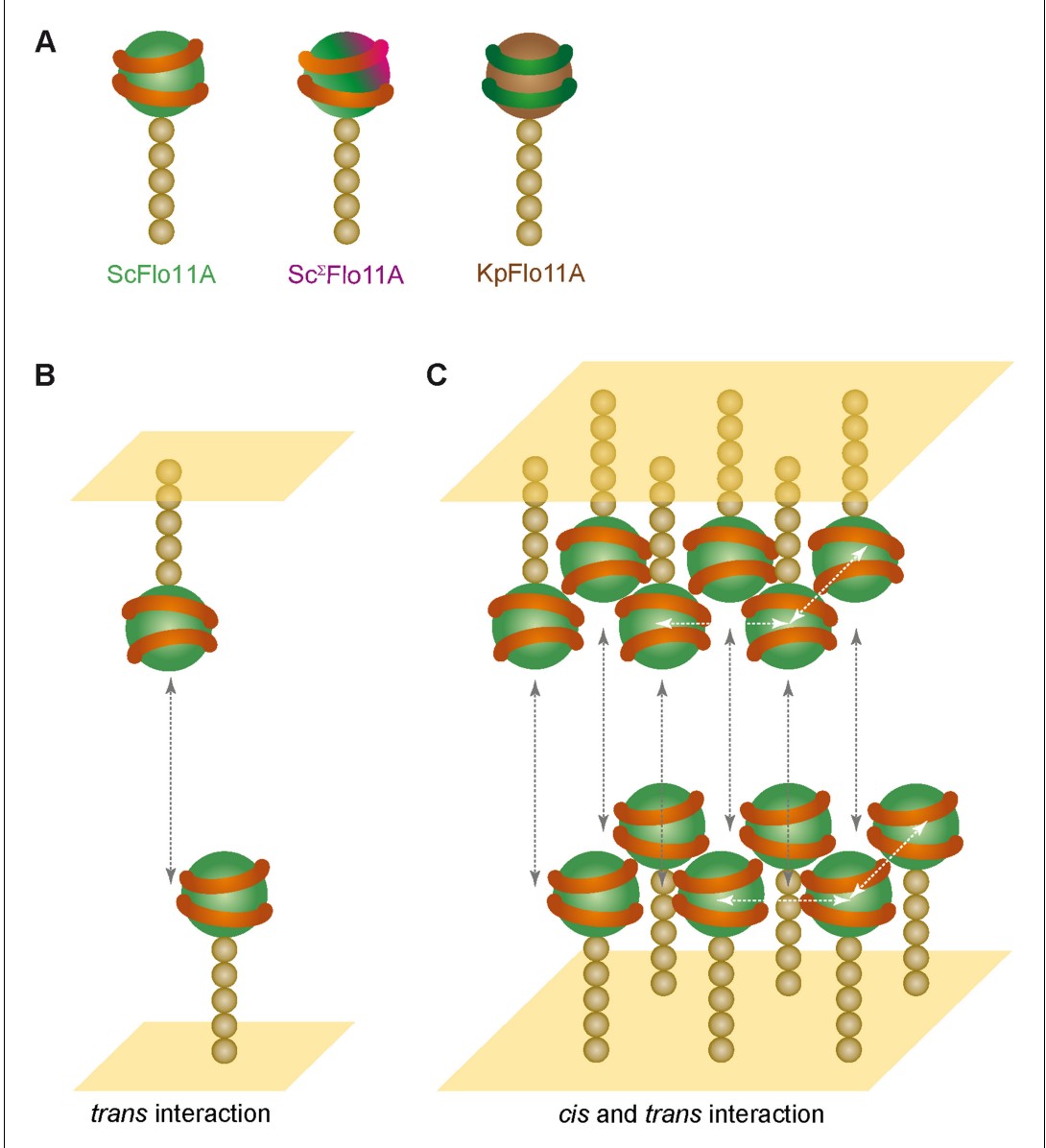

**Figure 8.** Model for kin discrimination by Flo11. (A) Structural elements for Flo11-directed discrimination. Shown are ScFlo11A, Sc$^\Sigma$Flo11A and KpFlo11A variants that differ either by the topology of the conserved aromatic bands shown as orange or green rings, respectively, or by an additional insertion shown in magenta. The Flo11BC domain is shown as beige spheres. (B) Model for bi-molecular Flo11A-Flo11A *trans*-interaction. Cell surfaces are shown in yellow and *trans*-interaction is indicated as grey, dotted arrow. (C) Model for interaction of Flo11A collectives by *cis*- and *trans*-interactions. Collective-forming *cis*-interactions are indicated as white, dotted arrows.

suggested that adhesion can act as an important weapon in microbial competition by demonstrating that highly adhesive bacterial cells can outcompete less adhesive cells in growing biofilms (*Schluter et al., 2015*). Our study also reveals that strong adhesion can correlate with dominance against less adhesive cells. This is exemplified by the Sc$^\Sigma$Flo11A variant, which in competitive assays dominates against all other Flo11 variants tested and confers roughly 2-fold higher cell-cell adhesion forces in comparison to its competitors. When comparing ScFlo11A with KpFlo11A, however, our study reveals a clear dominance of KpFlo11A, although ScFlo11A confers roughly 1.5-fold higher cell-cell adhesion forces. Moreover, both variants produce biofilms of comparable sizes indicating that cell-substrate adhesion forces are comparable. Thus, one might speculate that competition forces other than cell adhesion could be involved in conferring dominance, but such forces are difficult to conceive, because our experimental setup is based on yeast strains that genetically differ only

by the two Flo11 adhesion domains. We could rather imagine that the biophysical and adhesion properties of these two variants, which share only 32% identity, are differentially affected by changing microenvironments within growing biofilms. Unfortunately, however, quantification of individual cell-cell or cell-substrate adhesion forces in growing biofilms by AFM-based methods is currently not feasible.

Finally, our study has revealed that the function of Flo11-type proteins in conferring kin-directed cell-cell aggregation and discrimination can be observed in several families of Saccharomycetales. Thus, the importance of these adhesins for building multicellular consortia for kin-directed protection or other types of cooperative interaction appears to be conserved. Moreover, our study suggests that Flo11-directed kin discrimination could be accomplished at sub-species level by a single genetic event, as exemplified by the Σ-insert. Our finding that variants carrying this insert are present only in the 'Sake' lineage of non-mosaic *S. cerevisiae* strains, further opens the possibility that Σ-like sub-domains originate from the domestication of sake yeast strains. However, it remains to be elucidated whether the observed sub-species discrimination is indeed the result of a selective process.

# Materials and methods

## Key resources table

| Reagent type (species) or resource | Designation | Source or reference | Identifiers | Additional information |
|---|---|---|---|---|
| Gene (*Saccharomycetales*) | Flo11-type A domain sequences | various | | See *Supplementary file 1* |
| Genetic reagent *Saccharomyces cerevisiae* | Flo11-type A domain variants carrying yeast strains | | | See *Supplementary file 2* |
| Antibody | Rabbit polyclonal anti-ScFlo11A | (*Kraushaar et al., 2015*) | | Previously produced in our lab (1:10.000) |
| Antibody | Mouse monoclonal anti-STREP II Chromeo 488 | IBA Lifesciences, Göttingen, Germany | Cat# 2-1544-050 | (1:1.000) |
| Recombinant DNA reagent | Flo11-type A domain variants carrying plasmids | various | | See *Supplementary file 3* |
| Sequence-based reagent | Primer Ampl_FLO11_wt_fw | This study | PCR primers | CTC AAA AAT CCA TAT ACG CAC ACT |
| Sequence-based reagent | Primer Ampl_FLO11_wt_BC_rev | This study | PCR primers | GTA GAG CTG GAT GGA GTT GG |
| Peptide, recombinant protein | ScFlo11A | (*Kraushaar et al., 2015*) | | Previously produced in our lab |
| Peptide, recombinant protein | KpFlo11A | This study | | |
| Commercial assay or kit | Amine coupling kit | GE Healthcare, Solingen, Germany | | |
| Software, algorithm | Matlab | MathWorks | RRID:SCR_001622 | See section 'QCAM' and 'Competitive biofilm assay' |
| Software, algorithm | GraphPad Prism, v7.0 | GraphPad Software, La Jolla, CA, USA | RRID:SCR_002798 | |
| Software, algorithm | XDS | (*Kabsch, 2010*) | RRID:SCR_015652 | |
| Software, algorithm | CCP4 | *Collaborative Computational Project, Number 4, 1994* | RRID:SCR_007255 | |
| Software, algorithm | Phenix suite | (*Adams et al., 2002*) | RRID:SCR_014224 | |

*Continued on next page*

*Continued*

| Reagent type (species) or resource | Designation | Source or reference | Identifiers | Additional information |
|---|---|---|---|---|
| Software, algorithm | Coot | (*Emsley et al., 2010*) | RRID:SCR_014222 | |
| Software, algorithm | Refmac5 | (*Murshudov et al., 2011*) | RRID:SCR_014225 | |
| Software, algorithm | PyMOL v1.8.0.1 | (*Delano, 2002*) | RRID:SCR_000305 | |
| Software, algorithm | Biacore T100 evaluation software, v1.1 | GE Healthcare, Solingen, Germany | | |
| Software, algorithm | Clustal Omega web tool | (*McWilliam et al., 2013*) | RRID:SCR_001591 | |
| Software, algorithm | PRANK web tool | (*Löytynoja and Goldman, 2010*) | RRID:SCR_017228 | |
| Software, algorithm | FigTree v1.4.2 | http://tree.bio.ed.ac.uk | RRID:SCR_008515 | |
| Software, algorithm | STRIDE web server | (*Heinig and Frishman, 2004*) | | |
| Software, algorithm | Modeller 9v7 | (*Sali and Blundell, 1993*) | RRID:SCR_008395 | |

## Yeast strains and growth conditions

Yeast strains used in this study are shown in *Supplementary file 2*. For expression of plasmid-based *FLO11* genes, the non-adhesive *S. cerevisiae* strains RH2520 and RH2662 were used (*Grundmann et al., 2001*). To construct yeast strains YHUM3014, YHUM3015, YHUM3016, YHUM3017, YHUM3018, YHUM3019, YHUM3020, YHUM3021, YHUM3022 and YHUM3023 for competitive biofilm assays, the *flo11Δ* yeast strain RH2681 was labeled with RFP or GFP using integrative plasmids BHUM3349 and BHUM3350 respectively, followed by genomic integration of plasmid BHUM3353, BHUM3354, BHUM3356, BHUM3359 or BHUM3360. Yeast culture medium preparation and yeast transformations were conducted following standard protocols (*Guthrie and Fink, 1991*). For biofilm formation strains were grown on semisolid agar medium (*Reynolds and Fink, 2001*).

## Plasmids

Plasmids used in this study are listed in *Supplementary file 3*. Plasmids BHUM2223, BHUM2413, BHUM2419, BHUM2445, BHUM2753, BHUM2754, BHUM2756, BHUM2758, BHUM2883, BHUM2884, BHUM2906, BHUM2907, BHUM2920, BHUM2889, BHUM2890, BHUM2923, BHUM2939, BHUM2941, BHUM2955, BHUM2987, BHUM2989, BHUM3000 were constructed by insertion of appropriate *FLO11A* domains into expression plasmid BHUM2200 as previously described (*Kraushaar et al., 2015*). Plasmids BHUM3353, BHUM3354, BHUM3356, BHUM3359 and BHUM3360 were constructed using the SalI/BstAPI fragment from BHUM0778 containing $P_{FLO11\_\Sigma1278b}$ and the BstAPI/EcoRI fragment from either BHUM2422, BHUM2220, BHUM2223, BHUM2889 or BHUM2890 containing *FLO11* variants and YIplac211 as an integrative vector. Plasmid BHUM3349 was obtained using the SacI/XhoI fragment containing $P_{TDH3}$-*RFP* from pDS90 for subcloning in pRS304. For BHUM3350 the RFP in BHUM3349 were replaced with GFP from pYM12 by homologous recombination. *FLO11A* domains were generated by synthesis of codon-optimized DNA fragments (Thermo Fisher Scientific GeneArt, Regensburg, Germany) or by site-directed mutagenesis. To obtain BHUM3113, the KpFlo11A fragment was amplified by PCR using the genomic DNA of the *K. pastoris* strain GS115 and the primer KpFlo11-1_fw (5'-CGC G<u>CA TAT G</u>AG CTC AGG AAG ACT TGC CCT AC-3') and KpFlo11-1_rev (5'-CGC G<u>CT CGA G</u>TT ATG TAG TTG GTT CAA CAC CAC AGT CG-3') and subsequent insertion in pET28a(+).

## Recombinant overproduction of Flo11A proteins

Expression and purification of ScFlo11A and KpFlo11A domains were performed as previously described (*Kraushaar et al., 2015*).

## Isolation and analysis of *FLO11A* domains from *S. cerevisiae* strains

Novel *S. cerevisiae FLO11A* domains were isolated and analyzed by (i) PCR using chromosomal DNA isolated from appropriate yeast strains as template together with primers Ampl_FLO11_wt_fw (5'-C TC AAA AAT CCA TAT ACG CAC ACT-3') and Ampl_FLO11_wt_BC_rev (5'-GTA GAG CTG GAT GGA GTT GG-3'), (ii) cloning of the resulting DNA fragments using plasmid pJET1.2 (Thermo Fisher Scientific, Regensburg, Germany) and (iii) commercial DNA sequence analysis. All sequences are listed in *Supplementary file 1*.

## Adhesive growth assay

Adhesive growth of *S. cerevisiae* on agar surfaces was assayed as previously described (*Roberts and Fink, 1994*). At least four biological replicates of independently obtained clones were tested for every yeast strain.

## Quantitative cell aggregation microscopy (QCAM)

Strains from *S. cerevisiae* were grown on solid medium for 6 d before yeast cells from the center of the colonies with an area of 9 mm$^2$ were harvested. The cells were suspended in 10 ml liquid medium by vortexing for 5 s. Then, cellular aggregates were allowed to settle down for 30 s before aliquots of 70 µl were taken from the bottom of the test tube and spread on solid medium. After the residual liquid was drawn-in, images of cellular aggregates of a defined surface area (1 cm$^2$) were acquired using a Stemi 2000-C microscope (Zeiss, Jena, Germany) and the CMOS-camera Canon EOS1300D (Canon, Tokyo, Japan), with fixed exposure and aperture settings. Automated cell aggregate size analysis was performed using Matlab (MathWorks) as follows: (i) Images were subtracted from an averaged background image (surface of solid medium without cells) to correct inhomogeneous background and to invert the intensity scale yielding bright cell clumps. (ii) Cell clump segmentation was performed by applying a manually set global threshold, which was identical for all processed images. Obviously clumped cell aggregates were dissected by watershedding and false objects (i.e. dust) were manually removed. (iii) The cell aggregate size distribution was determined. At least two biological replicates of independently obtained clones and three technical replicates from each clone were tested.

## Competitive biofilm assay

Cultures of competing strains were grown to logarithmic phase before each approximately 500 cells were mixed in 3 µl liquid medium and spotted on the center of a plate containing semi-solid (0.2% agar) YEPD medium. After three weeks of incubation at room temperature, the plates were photographed using a Fuji Film LAS-4000 digital camera system using RFP and GFP filters. The resulting 8bit greyscale pictures were used for segmentation and quantification of the individual signals. The distribution of RFP and GFP signals was analyzed using the Matlab software (MathWorks) using he following procedure: (i) the colony edge was obtained by fusing the fluorescent signals into a gray level image and expanding the brightest region by iteratively using Otsu's threshold algorithm (*Otsu, 1979*); (ii) prior to the comparison, the fluorescent channels were normalized to discount for the auto-fluorescence of the medium and the different magnitudes in the GFP and RFP fluorescence; (iii) the relative abundance was given by the ratio of pixels that exhibit a RFP signal higher than the GFP signal within the whole colony as well as along a line in distance of 50 pixels to the colony edge to all pixels along the same line. For quantification of the genotypic ratio at the outer edge of biofilm, the amount of the signal produced by the superior strain was divided by the amount of the signal from the inferior strain. For each combination of competing strains, at least two biological replicates of independently obtained clones were assayed.

## Protein crystallization, data collection, and refinement

Initial crystallization attempts for KpFlo11A were performed with a Cartesian robot system and sparse-matrix screens (Qiagen, Hilden, Germany) in 96-well format using the sitting-drop vapor diffusion method. After 2 days, crystals of KpFlo11A-His$_6$ were obtained at 4°C using a protein concentration of 70 mg/ml in 20 mM HEPES buffer (pH 7.5) and 0.1 M sodium acetate, 0.2 M ammonium acetate pH 4.6, and 15% PEG4000 as precipitant as well as 0.1 M sodium acetate pH 4.6% and 12% PEG4000 as precipitant. Reproduction was carried out in 24-well format with drops containing 1 µl

of protein (35 mg/ml) and 1 µl of reservoir resolution equilibrated against 1 ml of reservoir. Cryoprotection of the crystals for flash freezing in liquid nitrogen was achieved by adding 30% glycerol to both conditions. X-ray data were collected from a single crystal at 100 K at the beamline MX-14–1, BESSY II, Helmholtz-Centre Berlin (HZB), Germany, with a Pilatus 6M detector. Two crystal forms could be obtained, the first one in space group $P2_12_12_1$ with unit cell parameters $a$ = 35.54 Å, $b$ = 58.52 Å, $c$ = 76.55 Å, and $\alpha = \beta = \gamma = 90°$ and one molecule per asymmetric unit (*Supplementary file 4*). The second one in space group $P2_1$ with unit cell parameters $a$ = 37.44 Å, $b$ = 58.68 Å, $c$ = 85.27 Å, and $\alpha = 90°$, $\beta = 96.56°$, $\gamma = 90°$ and two molecules per asymmetric unit. Data reduction was carried out using XDS, XSCALE, and SCALA from the CCP4 package (*Collaborative Computational Project, Number 4, 1994*). Both structures were solved by molecular replacement using the phenix suite (*Adams et al., 2002*) and structural data of a model from KpFlo11A that was created by Modeller 9v7 (*Sali and Blundell, 1993*) with ScFlo11A as template (PDB 4UYR). Further model building was done with Coot (*Emsley et al., 2010*), and automatic refinement with Refmac5 (*Murshudov et al., 2011*) and phenix.refine. Refinement of anisotropic thermal factors was used for 5FV5. All structures were visualized with PyMOL (*Delano, 2002*). Further details can be found in *Supplementary file 5*.

## SPR spectroscopy of ScFlo11A and KpFlo11A

Surface plasmon resonance was applied for the analysis of homo- and heterophilic interactions of KpFlo11A and ScFlo11A using a BiacoreT100 system (GE Healthcare, Solingen, Germany). KpFlo11A-His$_6$ was immobilized on a series S sensor chip CM5 via amine coupling in 10 mM acetate buffer (pH 4.5) with a concentration of 200 µg/ml and an amine coupling kit (GE Healthcare, Solingen, Germany). Immobilization was carried out at a flow rate of 10 µl/min and a contact time of 700 s on flow cell 2, while flow cell one was treated as blank immobilization and subtracted from all further measurements. KpFlo11A-His$_6$ was immobilized to a level of ~5,000 RU onto the chip surface. Binding experiments were performed in running buffer I (20 mM acetate buffer pH 5.5, 150 mM NaCl, 0.005% [v/v] surfactant P20) or running buffer II (20 mM HEPES buffer pH 7.5, 150 mM NaCl, 0.005% [v/v] surfactant P20). KpFlo11A-His$_6$ was used as analyte in concentrations from 0 to 270 µM, while the range for ScFlo11A was from 0 to 350 µM. Binding experiments were performed with a contact time of 360 s, flow rate of 25 µl/min, and dissociation time of 600 s. Regeneration was done for two cycles with regeneration buffer (100 mM TRIS-HCl pH 9.0; 2 M NaCl), contact time of 30 s, flow rate of 30 µl/min, and attaching stabilization period of 180 s. Results were analyzed with the Biacore T100 evaluation software, version 1.1.

## Preparation of single yeast cells for Atomic Force Microscopy (AFM)

Single cells suitable for measurement of yeast cell-cell adhesion by AFM were prepared by growth of yeast strains in SC-4 medium to a density of $5 \times 10^8$ cells per ml, followed by centrifugation and suspension of cells in YEPD medium. Clumped cells were then mechanically dissociated under a bench-top microscope (Nikon Instruments, Egg, Switzerland), using patch pipettes with a resistance of 10 MΩ mounted on a micromanipulator (Sutter Instrument, Novato, CA, USA). Patch pipettes were pulled on a P-97 Flaming/Brown pipette puller (Sutter Instrument, Novato, CA, USA) using Sutter glass capillaries with 1 mm outer and 0.5 mm inner diameter and subsequently flame-blunted using a Bunsen burner (*Figure 2—figure supplement 1*). Dissociated cells were again gently vortexed and plated on 35 mm glass bottom Petri dishes (World Precision Instruments, Sarasota, FL, USA) for subsequent single cell adhesion experiments.

## Single cell force spectroscopy (SCFS)

Single cell force spectroscopy (SCFS) was conducted with a CellHesion II Atomic Force Microscope (JPK Instruments, Berlin, Germany) mounted on an Axiovert 200 inverted fluorescence microscope (Zeiss, Jena, Germany) equipped with a 20x objective and used in closed height feedback mode. Differential interference contrast (DIC) imaging was used to monitor cellular morphology. Prior to use, each NPO-010 tip-less AFM cantilever (Bruker AFM Probes, Camarillo, CA, USA) was calibrated three times using thermal noise to eliminate errors. Spring constants were within 10% of the nominal value (~60 mN/m). Plasma-activated cantilevers were incubated with 2.5 mg/ml Concanavalin A (ConA; Sigma-Aldrich, Buchs, Switzerland) overnight at 4°C and carefully rinsed in phosphate

buffered saline (PBS) before use (*Schubert et al., 2014*). All yeast cell adhesion experiments were carried out in YEPD media on 35 mm glass bottom petri dishes (World Precision Instruments, Sarasota, FL, USA) carrying the appropriate target cell type. All SCFS measurements were carried out at 30℃ using a Petri dish heater (JPK Instruments, Berlin, Germany). Probe cells were obtained by gently pressing a ConA-coated cantilever onto appropriate cells and applying a force of approximately 2 nN for 3 s, followed by lifting the cantilever for 10 min to obtain firm cell attachment. For subsequent cell-cell adhesion experiments, probe cells on the cantilever were moved above target cells fixed by ConA on the surface of 35 mm glass bottom Petri dishes (*Kragl et al., 2016*). Cell-cell adhesion was measured by applying a contact force of 1 nN for 0.5 s to 20 s and employing approach and retraction velocities of 5 µm/s. For a given cell-cell couple, the contact time was varied randomly to prevent systematic bias or history effects. Each force-distance curve measuring the adhesion between probe and target cells was repeated several times: 1 s contact time, five repetitions; 10 s contact time, three repetitions; 20 s contact time, two repetitions. A resting time of 30 s was given between recording each force-distance curve. Individual probe cells were tested against multiple target-cells. One force-distance curve was determined with any given probe cell. Cells were continuously monitored during the SCFS experiment to assure intact and stable association with the cantilever and the substrate. Force-distance curves were analyzed using the JPK analysis software to extract maximum adhesion forces ($F_{max}$). Statistical analysis for calculation of average $F_{max}$ values at different time points was performed using the Prism software (GraphPad Software, La Jolla, CA, USA).

## Quantification of Flo11A proteins at the cell surface

Prior to adhesive growth assays, QCAM analysis and AFM measurements, the amount of Flo11A variants at the cell surface of yeast strains was quantified by immunofluorescence microscopy (*Supplementary file 6*) using polyclonal anti-ScFlo11A antibodies following protocols described previously (*Veelders et al., 2010*; *Kraushaar et al., 2015*) or a monoclonal anti-STREP II antibodies (Chromeo 488, IBA, Göttingen, Germany). For AFM analysis, ScFlo11A at the surface of probe and target cells was quantified after measurements by staining of cells under a cantilever applying a constant force of 2 nN. Stained cells were subsequently imaged on the AFM mounted on an Observer. Z1 LSM 700 inverted confocal microscope (Zeiss, Jena, Germany) at 30℃ using a Petri dish heater (JPK Instruments, Berlin, Germany). Within a given sample, all images were recorded using the same laser power and gain control. Images were acquired using a Zeiss 63x LCI Plan-Neofluar water immersion objective (NA 1.20). 2D maximum intensity projections were computed using the Imaris software (Bitplane, Zurich, Switzerland).

## Live/dead cell imaging for AFM

Single cells bound to the AFM cantilever were assessed for viability using the FUN-1 stain (Invitrogen, Carlsbad, USA). 2 µl of FUN-1 was added to the AFM adhesion measurement chamber immersed in YEPD media and incubated in the dark for 30 min. The cells were washed twice with YEPD media, and then imaged using a confocal microscope as shown previously (*Millard et al., 1997*). Live cells result in the formation of fluorescent red cylindrical intravacuolar structures (CIVS).

## Scanning electron microscopy

Pipettes were prepared for SEM imaging as previously described (*Schubert et al., 2018*). Briefly, they were fixed on a custom-made pipette holder and sputter-coated, with 4:1 gold:palladium mix for 60 s, using a Leica Ace200 low vacuum coater (Leica, Zurich, Switzerland). Pipettes were then imaged using a Zeiss 1550 field-emission scanning electron microscope at 3–5 kV.

## Chemical functionalization of cells

The cantilevers were washed with a 1:1 ratio of HCl and methanol (Sigma-Aldrich, Buchs, Switzerland). Cantilevers and glass bottomed WPI dishes were silanized with 5% 3-Aminopropyltriethoxysilane (Sigma-Aldrich, Buchs, Switzerland) in toluene for one hour. The amine ends of the cantilever were reacted with the NHS-ester of a heterobifunctional crosslinker, NHS-PEO12-maleimide (Pierce Scientific, ThermoFisher, Reinach, Switzerland), at a concentration of 2 mM crosslinker in PBS for 30 min as previously shown. Silanized glass-bottomed WPI dishes were passivated using amine-PEG-

hydroxyl (Sigma-Aldrich, Buchs, Switzerland) (*Morfill et al., 2007*; *Lee et al., 2010*; *Alsteens et al., 2017*). After reaction, the functionalized cantilevers were washed with PBS, the cantilever-crosslinker mix was then brought into contact with an isolated cell for 10 min in PBS. The reaction was then washed in YEPD medium and presented for adhesion measurement after 10 min of equilibration in YEPD medium.

## Bioinformatic and statistical analysis

Alignments of Flo11A protein sequences were generated using the Clustal Omega web tool (*McWilliam et al., 2013*). Molecular evolutionary genetic analysis of Flo11A sequences was performed by use of the PRANK web tool (*Löytynoja and Goldman, 2010*) and visualization of the results with the FigTree v1.4.2 open source software (http://tree.bio.ed.ac.uk). Known Flo11A sequences were obtained from the UniProt database (http://www.uniprot.org/) or published genome sequence data (*Liti et al., 2009*). For secondary structure assignment from known atomic coordinates of Flo11A proteins the STRIDE web server was used (*Heinig and Frishman, 2004*). To obtain a homology model of Flo11A from *S. cerevisiae* strain Σ1278b, the Modeller 9v7 software was used together with the template structure of Flo11A from *S. cerevisiae* strain S288c (4UYR) (*Kraushaar et al., 2015*). Due to conservation of a disulphide bridge between C128 from the Σ insert and C179 from the apical region intervening the β9 and β10 strands this initial model was subjected to 100 ns equilibration by molecular dynamics using the Amber14 suite and ff14sb force field parameters (*Maier et al., 2015*). Figures of protein structures were generated with the PyMOL v1.8.0.1 molecular graphics software (*Delano, 2002*). Statistical analysis of data obtained by SCFS measurements was performed using the Prism software (GraphPad Software, La Jolla, CA, USA) and for QCAM and biofilm analysis using Matlab (MathWorks). Statistical differences were considered as not significant with a P value > 0,05 and as significant with a P value < 0.05. p>0.05 (n.s), 0.05 ≥ P > 0.01 (*), 0.01 ≥ P > 0.001 (**), p≤0.001 (***).

## Acknowledgements

We thank Diana Kruhl for technical support, Sascha Liepelt and Alexander Reuß for helpful discussions and Christian von Wallbrunn for providing yeast stains.

## Additional information

### Funding

| Funder | Grant reference number | Author |
| --- | --- | --- |
| Deutsche Forschungsgemeinschaft | SFB 987 | Hans-Ulrich Mösch<br>Stefan Brückner<br>Timo Kraushaar<br>Raimo Hartmann<br>Eric Jelli<br>Daniel Hoffmann<br>Knut Drescher<br>Lars Oliver Essen, University of Marburg |
| Max-Planck-Gesellschaft | | Raimo Hartmann<br>Eric Jelli<br>Knut Drescher |

The funders had no role in study design, data collection and interpretation, or the decision to submit the work for publication.

### Author contributions

Stefan Brückner, Conceptualization, Formal analysis, Validation, Investigation, Visualization, Methodology, Writing - original draft, Writing - review and editing; Rajib Schubert, Timo Kraushaar, Raimo Hartmann, Formal analysis, Validation, Investigation, Visualization, Methodology, Writing - original draft, Writing - review and editing; Daniel Hoffmann, Investigation; Eric Jelli, Formal analysis, Validation, Investigation, Writing - original draft; Knut Drescher, Daniel J Müller, Supervision, Funding

acquisition, Validation, Writing - original draft, Writing - review and editing; Lars Oliver Essen, Conceptualization, Supervision, Funding acquisition, Validation, Visualization, Writing - original draft, Project administration, Writing - review and editing; Hans-Ulrich Mösch, Conceptualization, Formal analysis, Supervision, Funding acquisition, Validation, Visualization, Writing - original draft, Project administration, Writing - review and editing

### Author ORCIDs
Rajib Schubert (iD) https://orcid.org/0000-0002-7071-0134
Raimo Hartmann (iD) http://orcid.org/0000-0002-4924-6402
Knut Drescher (iD) http://orcid.org/0000-0002-7340-2444
Daniel J Müller (iD) https://orcid.org/0000-0003-3075-0665
Hans-Ulrich Mösch (iD) https://orcid.org/0000-0002-4660-6070

### Decision letter and Author response
Decision letter https://doi.org/10.7554/eLife.55587.sa1
Author response https://doi.org/10.7554/eLife.55587.sa2

## Additional files

### Supplementary files
- Supplementary file 1. Flo11A domain sequences.
- Supplementary file 2. Yeast strains.
- Supplementary file 3. Plasmids.
- Supplementary file 4. Crystal structure data collection, processing and refinement.
- Supplementary file 5. Structural analysis of KpFlo11A.
- Supplementary file 6. Quantification of Flo11A protein amounts.
- Transparent reporting form

### Data availability
Novel FLO11A DNA sequences have been deposited at the GenBank database under the consecutive accession numbers KX189102-KX189121. The atomic coordinates and structure factors have been deposited in the Protein Data Bank (www.rcsb.org) and assigned the accession codes 5FV5 and 5FV6.

The following datasets were generated:

| Author(s) | Year | Dataset title | Dataset URL | Database and Identifier |
|---|---|---|---|---|
| Brückner S, Mösch HU | 2016 | Saccharomyces cerevisiae strain SIHA_7 Flo11p (FLO11) gene, partial cds | https://www.ncbi.nlm.nih.gov/nuccore/KX189102 | NCBI GenBank, KX189102.1 |
| Brückner S, Mösch HU | 2016 | Saccharomyces cerevisiae strain SIHA_White_arome Flo11p (FLO11) gene, partial cds | https://www.ncbi.nlm.nih.gov/nuccore/KX189103 | NCBI GenBank, KX189103.1 |
| Brückner S, Mösch HU | 2016 | Saccharomyces cerevisiae strain Lalvin_R-HST Flo11p (FLO11) gene, partial cds | https://www.ncbi.nlm.nih.gov/nuccore/KX189104 | NCBI GenBank, KX189104.1 |
| Brückner S, Mösch HU | 2016 | Saccharomyces cerevisiae strain Uvaferm_SVG Flo11p (FLO11) gene, partial cds | https://www.ncbi.nlm.nih.gov/nuccore/KX189105 | NCBI GenBank, KX189105.1 |
| Brückner S, Mösch HU | 2016 | Saccharomyces cerevisiae strain Uvaferm_CEG Flo11p (FLO11) gene, partial cds | https://www.ncbi.nlm.nih.gov/nuccore/KX189106 | NCBI GenBank, KX189106.1 |
| Brückner S, Mösch HU | 2016 | Saccharomyces cerevisiae strain SSI2 Flo11p (FLO11) gene, partial cds | https://www.ncbi.nlm.nih.gov/nuccore/KX189107 | NCBI GenBank, KX189107.1 |

| Brückner S, Mösch HU | 2016 | Saccharomyces cerevisiae strain SSI6 Flo11p (FLO11) gene, partial cds | https://www.ncbi.nlm.nih.gov/nuccore/KX189108 | NCBI GenBank, KX189108.1 |
|---|---|---|---|---|
| Brückner S, Mösch HU | 2016 | Saccharomyces cerevisiae strain YJM128 Flo11p (FLO11) gene, partial cds | https://www.ncbi.nlm.nih.gov/nuccore/KX189109 | NCBI GenBank, KX189109.1 |
| Brückner S, Mösch HU | 2016 | Saccharomyces cerevisiae strain YJM222 Flo11p (FLO11) gene, partial cds | https://www.ncbi.nlm.nih.gov/nuccore/KX189110 | NCBI GenBank, KX189110.1 |
| Brückner S, Mösch HU | 2016 | Saccharomyces cerevisiae strain YJM308 Flo11p (FLO11) gene, partial cds | https://www.ncbi.nlm.nih.gov/nuccore/KX189111 | NCBI GenBank, KX189111.1 |
| Brückner S, Mösch HU | 2016 | Saccharomyces cerevisiae strain YJM309 Flo11p (FLO11) gene, partial cds | https://www.ncbi.nlm.nih.gov/nuccore/KX189112 | NCBI GenBank, KX189112.1 |
| Brückner S, Mösch HU | 2016 | Saccharomyces cerevisiae strain YJM311 Flo11p (FLO11) gene, partial cds | https://www.ncbi.nlm.nih.gov/nuccore/KX189113 | NCBI GenBank, KX189113.1 |
| Brückner S, Mösch HU | 2016 | Saccharomyces cerevisiae strain YJM312 Flo11p (FLO11) gene, partial cds | https://www.ncbi.nlm.nih.gov/nuccore/KX189114 | NCBI GenBank, KX189114.1 |
| Brückner S, Mösch HU | 2016 | Saccharomyces cerevisiae strain SSI3 Flo11p (FLO11) gene, partial cds | https://www.ncbi.nlm.nih.gov/nuccore/KX189115 | NCBI GenBank, KX189115.1 |
| Brückner S, Mösch HU | 2016 | Saccharomyces cerevisiae strain SSI4 Flo11p (FLO11) gene, partial cds | https://www.ncbi.nlm.nih.gov/nuccore/KX189116 | NCBI GenBank, KX189116.1 |
| Brückner S, Mösch HU | 2016 | Saccharomyces cerevisiae strain SSI9 Flo11p (FLO11) gene, partial cds | https://www.ncbi.nlm.nih.gov/nuccore/KX189117 | NCBI GenBank, KX189117.1 |
| Brückner S, Mösch HU | 2016 | Saccharomyces cerevisiae strain A6 Flo11p (FLO11) gene, partial cds | https://www.ncbi.nlm.nih.gov/nuccore/KX189118 | NCBI GenBank, KX189118.1 |
| Brückner S, Mösch HU | 2016 | Saccharomyces cerevisiae strain A18 Flo11p (FLO11) gene, partial cds | https://www.ncbi.nlm.nih.gov/nuccore/KX189119 | NCBI GenBank, KX189119.1 |
| Brückner S, Mösch HU | 2016 | Saccharomyces cerevisiae strain KVL012 Flo11p (FLO11) gene, partial cds | https://www.ncbi.nlm.nih.gov/nuccore/KX189120 | NCBI GenBank, KX189120.1 |
| Brückner S, Mösch HU | 2016 | Saccharomyces cerevisiae strain C1 Flo11p (FLO11) gene, partial cds | https://www.ncbi.nlm.nih.gov/nuccore/KX189121 | NCBI GenBank, KX189121.1 |
| Kraushaar T, Brückner S, Mikolaiski M, Schreiner F, Veelders M, Mösch HU, Essen LO | 2016 | KpFlo11 presents a novel member of the Flo11 family with a unique recognition pattern for homophilic interactions | https://www.rcsb.org/structure/5FV5 | RCSB Protein Data Bank, 5FV5 |
| Kraushaar T, Brückner S, Mikolaiski M, Schreiner F, Veelders M, Mösch HU, Essen LO | 2016 | KpFlo11 presents a novel member of the Flo11 family with a unique recognition pattern for homophilic interactions | https://www.rcsb.org/structure/5FV6 | RCSB Protein Data Bank, 5FV6 |

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
