## [Decision Letter]

**Acceptance summary:**

Kin discrimination is broadly defined as differential treatment of conspecifics according to their relatedness. Kin discrimination could help biological systems to direct cooperative behaviour towards their relatives. In bacteria, social amoebae and yeast, cell surface adhesins have been suggested to confer kin discrimination by mediating adhesion between cells using either homotypic or heterotypic interactions, but the precise molecular and structural basis underlying the evolution of kin discrimination is largely unexplored.

In this manuscript, the authors discovered a molecular mechanism how social yeasts can discriminate each other. This mechanism is based on the Flo11-type adhesins that can be highly diverse. They demonstrate that in vivo Flo11-type adhesion domains from different or even the same yeast species confer efficient cell-cell adhesion at single cell and population level in homotypic configurations and are able to efficiently discriminate between kin and non-kin variants in heterotypic situations.

The manuscript reports novel aspects and in particular describes the discovery of a novel mechanism of kin discrimination in eukaryotes; in fact a novel kin discrimination molecule was discovered. The findings are of major relevance for many biological systems.

**Decision letter after peer review:**

[Editors’ note: the authors submitted for reconsideration following the decision after peer review. What follows is the decision letter after the first round of review.]

Thank you for submitting your work entitled "Kin discrimination in social yeast is mediated by cell surface receptors of the Flo11 adhesin family" for consideration by *eLife*. Your article has been reviewed by three peer reviewers, one of whom is a member of our Board of Reviewing Editors, and the evaluation has been overseen by a Senior Editor. The reviewers all opted to remain anonymous.

Our decision has been reached after consultation between the reviewers. Based on these discussions and the individual reviews below, we regret to inform you that your work will not be considered further for publication in *eLife* at this stage. However, we would be happy to review a new submission given the additional required experiments have been performed.

In general, the findings are very interesting and allow for novel insight in the mechanisms of kin discrimination in eukaryotes; in fact a novel kin discrimination molecule was discovered. The findings were obtained by the use of sophisticated methods like single cell force microscopy. However, we also identified some issues in your work before publication of your work by *eLife* can be considered. To address these issues additional experiments need to be performed. They concern:

1) The number of Flo11A molecules present on single cells (see reviewer 1).

2) Isogenic strains with different Flo11 alleles should be mixed together and tested whether discrimination actually occurs, i.e. the strains partition into distinct social groups.

3) Western blot analysis are needed to confirm that the Flo11A mutant phenotypes (e.g. Figure 2B and C) are not caused by unstable mutant proteins.

4) The study does not go far enough in parts to support the central hypothesis put forward in the title and in many other places in the paper that speak to "Kin discrimination in social yeast". The authors may be able to address this issue by setting up corresponding assays to measure cellular interactions between strains expressing different Flo11 versions in vivo and also by establishing competition assays. Such information will complement the strong biochemical and biophysical data and allow the authors to properly interpret aspects of microbial interactions.

5) The F_max_ required to separate cells with or without the Flo11A domain shows a range of 2 to 15 nN. Though the difference is shown to be statistically significant, there is little to suggest that it is also biologically relevant. What natural situations would a yeast find itself in where a force difference of 13 nN would matter to it? What natural environments do yeast exist in between 2 and 15 nN where this difference in affinity would presumably be relevant? Additionally, the authors attempt to categorize the experimental system as "in vivo" though the natural context of this type of interaction is not clear. An identical experiment could have been performed with immobilized Flo11p, yielding the same results, which is also not considered "in vivo."

6) The phenomenon of kin discrimination in vivo implies that there exists a certain benefit to the kin, but there is no exploration of this. There are no competition assays. There are no measurements of fitness as they relate to the relative strength of Flo11A interactions. Do *K. pastoris* form more vulnerable biofilms because they have weaker homotypic interactions than *S. cerevisiae*? Will a mixed population of *S. cerevisiae* and *K. pastoris* show *S. cerevisiae* as dominant over time?

7) Might the relatively weak hetertypic interactions be strong enough to hold cells together in the wild? The homotypic interactions and KpFlo11A are shown to be half as strong as homotypic interactions in ScFlo11A, however the authors also say this does affect the cell-cell aggregation assays.

8) Will cells expressing different Flo11A domains segregate within a biofilm?

9) Will cells expressing different Flo11A domains form separate flocs?

Reviewer #1:

Kin discrimination, broadly defined as differential treatment of conspecifics according to their relatedness, could help biological systems direct cooperative behaviour toward their relatives. In bacteria, social amoebae and yeast, cell surface adhesins have been suggested to confer kin discrimination by mediating adhesion between cells using either homotypic or heterotypic interactions, but the precise molecular and structural basis underlying the evolution of kin discrimination is largely unexplored.

In the current manuscript, the authors provide evidence that the Flo11-type adhesins of diverse social yeasts define kin discrimination. The authors demonstrate that in vivo Flo11-type adhesion domains from different or even the same yeast species confer efficient cell-cell adhesion at single cell and population level in homotypic configurations and are able to efficiently discriminate between kin and non-kin variants in heterotypic situations. Structural data based on x-ray and biochemical analyses show that the core and aromatic surface patterns of highly divergent Flo11A domains are similar, but not identical.

I was wondering how much the current data advance the field compared to the previous paper (Structure, 2015) of the group where the crystal structure of the N-terminal Flo11A domain was already determined. I feel that the current manuscript includes novel aspects and in particular also describes very interesting novel findings concerning the kin discrimination in a fungus.

In general, the findings are very interesting and give novel insight in the mechanisms of kin discrimination in eukaryotes; in fact a novel kin discrimination molecule was discovered. The findings were obtained by the use of sophisticated methods like single cell force microscopy. I believe the manuscript describes novel, far-reaching findings.

Although the measurements of the adhesion between single cells is convincing, an important factor was neglected. How many Flo11A molecules are present on the surface of the different yeasts? Also, what is the number of recombinant proteins on the surface of yeast expressing Flo11A derivatives on their surface? In my opinion, the adhesion force depends on hydrogen bonds but also on the number of surface molecules providing these hydrogen bonds. It cannot be excluded that the kin recognition depends more on the number of surface molecules that might be different in the different yeast strains rather than the direct adhesion force per molecule. Because the difference in adhesion force between homophilic and heterophilic configurations is at best a factor of 10, this point needs further experimental clarification, e.g., by labeling Flo11A with GFP and measuring the overall fluorescence intensity.

Reviewer #2:

In this study, Brückner and co-authors investigated how yeast use the Flo11-type adhesin to mediate kin discrimination through homotypic interactions. There are several major findings including:

i) A reinforced and refined model of Flo11 homotypic interactions built from their prior work (Kraushaar et al., 2015), which incorporates new approaches, such as SCFS and QCAM. These approaches can be widely used to study other adhesins and biofilm-forming microbes.

ii) Through experimental analysis of Flo11A from 4 yeast strains that belong to 3 different species, they suggest that the Flo11A domains mediate selective cell-cell adhesion in social yeasts.

iii) They solved the crystal structure of a divergent Flo11A allele from *K. pastoris* to 1.4 angstroms and show that it's structure very similar to Flo11A from *S. cerevisiae*.

Overall, the work is nicely done and presented. However, one major shortcoming is that this study did not fully explain how discrimination between Flo11A alleles is achieved at the molecular/structural level, a key question they asked (Introduction). Nevertheless, they do show that aromatic bands I and II along with a 15 AA insert play important roles in homotypic discrimination between receptors. How this insert provides specificity and how prevalent it is in nature remains unclear.

The authors describe Flo11 as a greenbeard gene involved in kin discrimination, but they do not directly show this. To do so, isogeneic strains with different Flo11 alleles should be mixed together and tested whether discrimination actually occurs, i.e. the strains partition into distinct social groups.

Discussion section. The authors argue that Flo11 forms cell surface clusters that are driven by cis interactions between receptors. If this is true then clusters of Flo11 should be seen by IF, but instead the receptors are found to be evenly distributed around the cell surface (Figure 2—figure supplement B). This point should be considered and explained.

Western blot analysis are needed to confirm that the Flo11A mutant phenotypes (e.g. Figure 3B and C) are not caused by unstable mutant proteins.

Reviewer #3:

Cell adhesion is critically important for microbial cells to form biofilms/mats, to cooperate, and to undergo a wide variety of specialized behaviors. Understanding how cell adhesion leads to regulated interactions among microbial cells can provide insight into 'social' responses among individuals and the overall regulation of microbial responses. It has previously been shown in *Dictyostelium* and yeast that cell adhesion molecules can act as 'green beard genes' that foster interactions among like individuals (kin discrimination) through homotypic interactions. Furthermore, such interactions among cells in the exterior of a biofilm can protect cells in the interior from toxins.

The study identifies the region and critical residues on Flo11p that confer the attribute of kin discrimination to various yeast cells. The authors showed that: (1) the Flo11A domain is necessary for homotypic interactions and (2) various A domains from Flo11p homologs have stronger homotypic interactions than heterotypic interactions with different A domains. The authors also identify the critical aromatic residues within the Flo11A domain that are critical for aggregation. The approaches are novel (force microscopy) and robust (protein crystallography), and the findings have merit. However, the study does not go far enough in parts to support the central hypothesis put forward in the title and in many other places in the paper that speak to "Kin discrimination in social yeast". The authors may be able to address this issue by setting up corresponding assays to measure cellular interactions between strains expressing different Flo11 versions in vivo and also by establishing competition assays. Such information will complement the strong biochemical and biophysical data and allow the authors to properly interpret aspects of microbial interactions. More specific concerns and questions follow below:

1) Assays like the force measurement technique, which is quite novel and which was used to quantitate the strength of interactions, does not in itself demonstrate social interactions. This leads to problems. For example, the F_max_ required to separate cells with or without the Flo11A domain shows a range of 2 to 15 nN. Though the difference is shown to be statistically significant, there is little to suggest that it is also biologically relevant. What natural situations would a yeast find itself in where a force difference of 13 nN would matter to it? What natural environments do yeast exist in between 2 and 15 nN where this difference in affinity would presumably be relevant? Additionally, the authors attempt to categorize the experimental system as "in vivo" though the natural context of this type of interaction is not clear. An identical experiment could have been performed with immobilized Flo11p, yielding the same results, which is also not considered "in vivo."

2) The phenomenon of kin discrimination in vivo implies that there exists a certain benefit to the kin, but there is no exploration of this. There are no competition assays. There are no measurements of fitness as they relate to the relative strength of Flo11A interactions. Do K. pastoris form more vulnerable biofilms because they have weaker homotypic interactions than *S. cerevisiae*? Will a mixed population of *S. cerevisiae* and *K. pastoris* show *S. cerevisiae* as dominant over time?

3) Might the relatively weak hetertypic interactions be strong enough to hold cells together in the wild? The homotypic interactions and KpFlo11A are shown to be half as strong as homotypic interactions in ScFlo11A, however the authors also say this does affect the cell-cell aggregation assays.

4) The authors argue that differences in affinity among Flo11A domains allow cells to discriminate, but they do not show this in anything resembling a natural context.

5) Will cells expressing different Flo11A domains segregate within a biofilm?

6) Will cells expressing different Flo11A domains form separate flocs?

[Editors’ note: further revisions were suggested prior to acceptance, as described below.]

Thank you for submitting your article "Kin discrimination in social yeast is mediated by cell surface receptors of the Flo11 adhesin family" for consideration by *eLife*. Your article has been reviewed by three peer reviewers, one of whom is a member of our Board of Reviewing Editors, and the evaluation has been overseen Naama Barkai as the Senior Editor. The reviewers have opted to remain anonymous.

The reviewers have discussed the reviews with one another and the Reviewing Editor has drafted this decision to help you prepare a revised submission. The reviewers agree with each other that you have submitted a strong manuscript describing very interesting findings. This resubmitted version has improved considerably. However, there remain some questions that we ask you to address.

Essential revisions:

1) The interpretation of the results of the biofilm experiments is not clear. Many papers in yeast and bacteria have explored the topic using fluorescently labeled cells, and it is unclear whether you are taking all this information into account when interpreting your data. For example, in Figure 5, the biofilms have different sizes. Please take biofilm size into account for your analysis.

2) Measuring the fluorescence at the outer edge might not be the best way to measure success in biofilm populations. For example, cells 'trapped' in the middle may have an advantage being protected from the external environment. How about measuring the ratio of total fluorescence of the two signals.

*Reviewer #1:*

The manuscript is a resubmission of a previously submitted manuscript (2018) that was. It deals with Kin discrimination, broadly defined as differential treatment of conspecifics according to their relatedness. Kin discrimination could help biological systems to direct cooperative behaviour towards their relatives. In general, the question addressed is of broad relevance for all species.

In bacteria, social amoebae and yeast, cell surface adhesins have been suggested to confer kin discrimination by mediating adhesion between cells using either homotypic or heterotypic interactions, but the precise molecular and structural basis underlying the evolution of kin discrimination is largely unexplored.

In the current manuscript, the authors provide evidence that the Flo11-type adhesins of diverse social yeasts define kin discrimination. The authors demonstrate that in vivo Flo11-type adhesion domains from different or even the same yeast species confer efficient cell-cell adhesion at single cell and population level in homotypic configurations and are able to efficiently discriminate between kin and non-kin variants in heterotypic situations. Structural data based on x-ray and biochemical analyses show that the core and aromatic surface patterns of highly divergent Flo11A domains are similar, but not identical.

The current manuscript is very well written, the figures are of high quality. The manuscript includes novel aspects and in particular also describes very interesting novel findings concerning the kin discrimination in a fungus. As I wrote in my previous review, in general the findings are very interesting and give novel insight in the mechanisms of kin discrimination in eukaryotes; in fact, a novel kin discrimination molecule was discovered. The findings were obtained by the use of sophisticated methods like single cell force microscopy. I believe the manuscript describes novel, far-reaching findings.

The questions raised by myself in the last version of the manuscript were all well answered experimentally.

Specifically, the authors determined the number of Flo11A molecules present on the surface of the different yeast strains. This also included determination of the number of recombinant proteins on the surface of strains expressing Flo11A derivatives on their surface. The difference in numbers was less than 20%. Therefore, the conclusion of the authors based on differences in adhesion force (differences up to a factor of 10) between homophilic and heterophilic configurations is justified.

Also, the authors now provided in vivo competition assays addressing the behaviour of mixed populations expressing heterogenous Flo11 variants. For this purpose, the authors generated a set of isogenic yeast strains expressing Flo11 variants from different species/subspecies and different fluorescent markers (GFP or RFP) for genotypic tracking. This novel data was included as a new Figure 5 and shows significant differences between homotypic (single Flo11A variant) and heterotypic (two different Flo11A variants) biofilms with respect to the ratio of the Flo11A expressing strains at the outer edge of the growing biofilms.

*Reviewer #2:*

In their revised manuscript Bruckner et al. have done a thorough job addressing prior reviewer concerns. In particular they developed a new fitness assay to investigate the role of the Flo11 cell surface receptor in kin recognition. This assay clearly shows that homotypic interactions between receptors provides a fitness advantage for growth at the leading edge of an expanding colony. With isogenic strains, the assay also shows that particular Flo11 alleles have stronger fitness gains than other. As I commented in my prior review, this is a very nice study that provides structural and mechanistic insight into kin recognition in microbes. I have no other concerns.

*Reviewer #3:*

Understanding cell adhesion is important for the study of many aspects of biological systems. Cell adhesion in microbial species is important for social organization, and in pathogens, virulence. Pioneering discoveries in yeast, *Dictyostelium*, and other organisms have shown that homotypic interactions of adhesion molecules promote social cooperation. In yeast, Flo11 is the major adhesion molecule and a representative adhesion protein to study fungal cell adhesion, particularly during invasive growth and biofilm formation in yeast. Pioneering studies by the Fink lab, and more recently biochemical and biophysical studies by the Mosch lab, have defined the structural basis of adhesion of this molecule. In this study, the authors develop new tools to study Flo11 adhesion, including exploring the differences of Flo11 across species, measurement of adhesion forces using single-cell force spectroscopy (SCFS), and quantitative cell aggregation microscopy (QCAM). They conclude that changes in the Flo11 protein in part lead to refinement of social cooperation within species and among closely related species. Overall, this is an important study and would be appropriate for *eLife*.

The authors demonstrate that Flo11's A domain from different species can function in adhesion. They go on to show that Flo11's A domain promotes species-specific adhesion across several species. This result involved obtaining the crystal structure of Flo11 from Kp and comparing it to their previous structure in Sc. The authors go on to explore other features of the Flo11 protein, including an insertion loop, and develop two-fluorescent biofilm assays to examine competition between strains expressing Flo11 variants. The study is an interesting intersection between structural biology and evolutionary/competition biology. There are some issues that should be addressed below, but this is certainly an important advance above what is known. Although one could have easily speculated the outcome of the study, it is the first demonstration of species and subspecies discrimination by specific changes in a cell adhesion molecule.

The interpretation of the results of the biofilm experiments is not clear and may not be correct. Many papers in yeast and bacteria have explored the topic using fluorescently labeled cells, and I am not convinced that the authors are taking all this information into account when interpreting their data. For example, in Figure 5, the biofilms are different sizes. The authors should take biofilm size be taken into account for their analysis.

It is also not clear that measuring the fluorescence at the outer edge is the best way to measure success in biofilm populations. For example, cells 'trapped' in the middle may have an advantage being protected from the external environment.

It is not clear that more cells means that one cell type 'wins' over another?

How about measuring the ratio of total fluorescence of the two signals?

It is also not clear why ClFlo11 beats ScFlo11. Does this correspond to the relative adhesive forces of these molecules? As is, this figure could be inflammatory to the community of scientists who study social cooperation.

Why are cells not seen to mix into conglomerates of self organizing communities, as reported in many papers?

Might other competition forces shape the patterns observed here?

---

## [Author Response]

[Editors’ note: the authors resubmitted a revised version of the paper for consideration. What follows is the authors’ response to the first round of review.]

In general, the findings are very interesting and allow for novel insight in the mechanisms of kin discrimination in eukaryotes; in fact, a novel kin discrimination molecule was discovered. The findings were obtained by the use of sophisticated methods like single cell force microscopy. However, we also identified some issues in your work before publication of your work by eLife can be considered. To address these issues additional experiments need to be performed. They concern:1) The number of Flo11A molecules present on single cells (see reviewer 1).

In order to measure Flo11 molecules present at the cell surface, we routinely perform quantification by immunofluorescence microscopy, because this method is a good measure for the Flo11 molecules accessible for protein-protein interactions at the cell surface. Quantitative Western blot analysis is not feasible, because Flo11 proteins are highly glycosylated and do not reproducibly enter separation gels. In the previous version of our manuscript, we have only briefly described this quantification (Materials and methods section). We now describe the method in more detail and we have added the data as a new Supplementary file 5. Importantly, this data shows that all Flo11 variants used in the study are present at comparable amounts at the cell surface. Moreover, detectable protein amounts and functionality (as assayed by QCAM and agar adhesion) for the variants tested do not correlate, as for instance exemplified by the functional variant Y111A Y113A Y118A (86% relative surface amount) and the non-functional variant Y111D Y113D Y118D (92% relative surface amount).

2) Isogenic strains with different Flo11 alleles should be mixed together and tested whether discrimination actually occurs, i.e. the strains partition into distinct social groups.

We have now performed in vivo competition assays to address the behavior of mixed populations expressing heterogenous Flo11 variants. In order to address this issue, we have constructed a set of isogenic yeast strains expressing Flo11 variants from different species/subspecies (fitting the competitive AFM assays) and different fluorescent markers (GFP or RFP) for genotypic tracking. These strains were used for competitive biofilm assays, in order to measure Flo11-dependent segregation at population level in vivo. This novel data is included as new Figure 5 and clearly shows significant differences between homotypic (single Flo11A variant) and heterotypic (two different Flo11A variants) biofilms with respect to the ratio of the Flo11A expressing strains at the outer edge of the growing biofilms. Importantly, this ratio was found to be very low in homotypic biofilms, but significantly higher (up to more than 1000-fold) in heterotypic biofilms using strains expressing different Flo11A variants. This demonstrates that in competitive situations one Flo11 allele outcompetes the other and monopolizes the outer edge of the expanding biofilm. This novel data is in agreement with our data obtained by competitive AFM assays at single cell level and supports the conclusion that heterogenous Flo11A alleles are able to discriminate against each other at population level.

3) Western blot analysis are needed to confirm that the Flo11A mutant phenotypes (e.g. Figure 2B and C) are not caused by unstable mutant proteins.

See answer to point 1.

4) The study does not go far enough in parts to support the central hypothesis put forward in the title and in many other places in the paper that speak to "Kin discrimination in social yeast". The authors may be able to address this issue by setting up corresponding assays to measure cellular interactions between strains expressing different Flo11 versions in vivo and also by establishing competition assays. Such information will complement the strong biochemical and biophysical data and allow the authors to properly interpret aspects of microbial interactions.

See answer to point 2.

5) The F_max_ required to separate cells with or without the Flo11A domain shows a range of 2 to 15 nN. Though the difference is shown to be statistically significant, there is little to suggest that it is also biologically relevant. What natural situations would a yeast find itself in where a force difference of 13 nN would matter to it? What natural environments do yeast exist in between 2 and 15 nN where this difference in affinity would presumably be relevant? Additionally, the authors attempt to categorize the experimental system as "in vivo" though the natural context of this type of interaction is not clear. An identical experiment could have been performed with immobilized Flo11p, yielding the same results, which is also not considered "in vivo."

See answer to point 2.

With respect to a possible natural situation, biofilm formation on semi-solid surfaces (as conducted by us now) might reflect growth of yeast on naturally viscous environments, such as rotting fruits.

Regarding the difference between our QCAM assays using living whole cells and an aggregation assay using immobilized Flo11p, we think that there might be differences with respect to the local concentration and presentation of Flo11 molecules, which could be crucial for efficient trans interaction (see our Discussion section). However, we now refer to our QCAM assay in the improved manuscript as cell aggregation assay using whole populations.

6) The phenomenon of kin discrimination in vivo implies that there exists a certain benefit to the kin, but there is no exploration of this. There are no competition assays. There are no measurements of fitness as they relate to the relative strength of Flo11A interactions. Do K. pastoris form more vulnerable biofilms because they have weaker homotypic interactions than S. cerevisiae? Will a mixed population of S. cerevisiae and K. pastoris show S. cerevisiae as dominant over time?

See answer to point 2. Furthermore, it is interestingly to note that in our competitive biofilm assays, strains expressing *K. pastoris* Flo11 become slightly dominant over strains with *S. cerevisiae* Flo11. However, whereas AFM measures Flo11 interactions after 20 s (contact times exceeding 20 s could not be reliably assayed, because the probe cells were repeatedly torn from the cantilever), our competitive biofilm assays measure segregation over a much longer time period of 3 weeks. Unfortunately, we cannot measure individual cell-cell interactions by AFM at later time points.

7) Might the relatively weak hetertypic interactions be strong enough to hold cells together in the wild? The homotypic interactions and KpFlo11A are shown to be half as strong as homotypic interactions in ScFlo11A, however the authors also say this does affect the cell-cell aggregation assays.

See answer to point 2.

8) Will cells expressing different Flo11A domains segregate within a biofilm?

See answer to point 2.

9) Will cells expressing different Flo11A domains form separate flocs?

We have not addressed this question experimentally, but chosen to perform competition assays in biofilms, in order to measure separation over a long time period.

[Editors’ note: what follows is the authors’ response to the second round of review.]

Reviewer #3:1) The interpretation of the results of the biofilm experiments is not clear. Many papers in yeast and bacteria have explored the topic using fluorescently labeled cells, and it is unclear whether you are taking all this information into account when interpreting your data. For example, in Figure 5, the biofilms have different sizes. Please take biofilm size into account for your analysis.

We thank the reviewer for pointing out this gap. To address this issue, we have further analyzed all biofilms with regard to the biofilm sizes and the spatial distribution of competing yeast strains within the whole biofilm. This data is included as a new figure (Figure 5—figure supplement 1), which shows (A) the red/green signal ratios in relation to the distance from the center (inoculation spot) of the biofilms, (B) the biofilm size for each combination of competing yeast strains, (C) the relative presence of competing strains within whole biofilms, and (D) the relative presence of competing strains at the outer biofilm edge (previously presented data). The new data shows that in all heterotypic biofilms one of the alleles clearly dominates the biofilm with respect to its relative presence within the whole area covered. In addition, the total sizes of mixed biofilms generally correspond to the sizes of the homogeneous biofilms of dominant alleles. As such, the new data supports our previous conclusion that heterogenous Flo11A alleles are able to discriminate against each other in mixed populations. We further discuss the new data in an additional paragraph with respect to the question of how cell surface presentation and interaction of Flo11-type proteins translate into the development of multicellular growth forms under non-competitive and competitive conditions.

2) Measuring the fluorescence at the outer edge might not be the best way to measure success in biofilm populations. For example, cells 'trapped' in the middle may have an advantage being protected from the external environment. How about measuring the ratio of total fluorescence of the two signals.As stated above, we have measured the ratio of the total fluorescence of the two signals. This data supports the data measuring the fluorescence at the outer edge, because we observe that in all competitive combinations assayed one strain becomes dominant with respect to the relative area 'conquered' within the whole biofilm as a measure of success. Whether the cells 'trapped' in the middle are protected against the external environment cannot be directly concluded from our data and would have to be analyzed in a separate study analyzing an array of different harmful environmental conditions. Nevertheless, our data supports the main focus of this study concerning the ability of Flo11 adhesins to confer discrimination not only at protein and at single cell level, but also in heterogeneous populations.3) It is also not clear why ClFlo11 beats ScFlo11. Does this correspond to the relative adhesive forces of these molecules? As is, this figure could be inflammatory to the community of scientists who study social cooperation.

The reviewer is right. We also feel that this finding is intriguing, especially because we find that ScFlo11A confers roughly 1.4-fold higher cell-cell adhesion forces as measured by AFM, but on average leads to 1.8-fold lower biofilm sizes, when compared to ClFlo11A. This apparent discrepancy could be explained by postulating significant differences in cell-substrate adhesion conferred by these Flo11A variants. In other words, ClFlo11A might be able to interact with the substrate much more efficiently than ScFlo11A and thereby confer a significantly better spreading of the growing biofilm. It remains to be elucidated, whether and how different Flo11A variants confer adhesion to different substrate surfaces. We discuss these possibilities in a new paragraph of the revised manuscript.

4) Why are cells not seen to mix into conglomerates of self organizing communities, as reported in many papers?

We also think that studying the connection between intercellular adhesion and the internal structure of microbial colonies and biofilms at single cell level is an important topic. However, addressing such issues with regard to the Flo11 adhesin family would require a large body of additional experimental work such as a combination of high-resolution time-lapse microscopy and mathematical modelling, which in our opinion would clearly be beyond the scope of the current manuscript.

5) Might other competition forces shape the patterns observed here?

See answer to point 3.